# FASTER DIFFUSION SAMPLING WITH RANDOMIZED MIDPOINTS: SEQUENTIAL AND PARALLEL

**Shivam Gupta**[1]**, Linda Cai**[2]**, Sitan Chen**[3]
[1]UT Austin    [2]UC Berkeley    [3]Harvard SEAS
`shivamgupta@utexas.edu, tcai@berkeley.edu, sitan@seas.harvard.edu`

## ABSTRACT

Sampling algorithms play an important role in controlling the quality and runtime of diffusion model inference. In recent years, a number of works (Chen et al., 2023c;b; Benton et al., 2023; Lee et al., 2022) have analyzed algorithms for diffusion sampling with provable guarantees; these works show that for essentially any data distribution, one can approximately sample in polynomial time given a sufficiently accurate estimate of its score functions at different noise levels.

In this work, we propose a new scheme inspired by Shen and Lee's randomized midpoint method for log-concave sampling (Shen & Lee, 2019). We prove that this approach achieves the best known dimension dependence for sampling from arbitrary smooth distributions in total variation distance ($\widetilde{O}(d^{5/12})$ compared to $\widetilde{O}(\sqrt{d})$ from prior work). We also show that our algorithm can be parallelized to run in only $\widetilde{O}(\log^2 d)$ parallel rounds, constituting the first provable guarantees for parallel sampling with diffusion models.

As a byproduct of our methods, for the well-studied problem of log-concave sampling in total variation distance, we give an algorithm and simple analysis achieving dimension dependence $\widetilde{O}(d^{5/12})$ compared to $\widetilde{O}(\sqrt{d})$ from prior work.

## 1 INTRODUCTION

Diffusion models (Sohl-Dickstein et al., 2015; Song & Ermon, 2019; Ho et al., 2020; Dhariwal & Nichol, 2021; Song et al., 2021a;b; Vahdat et al., 2021) have emerged as the *de facto* approach to generative modeling across a range of data modalities like images (Betker et al., 2023; Esser et al., 2024), audio (Kong et al., 2020), video (Brooks et al., 2024), and molecules (Wu et al., 2024). In recent years a slew of theoretical works have established surprisingly general convergence guarantees for this method (Chen et al., 2023c; Lee et al., 2023; Chen et al., 2023a;b;d; Benton et al., 2024; Gupta et al., 2023b; Li et al., 2023; 2024). They show that for essentially any data distribution, assuming one has a sufficiently accurate estimate for its *score function*, one can approximately sample from it in polynomial time.

While these results offer some theoretical justification for the empirical successes of diffusion models, the upper bounds they furnish for the number of iterations needed to generate a single sample are quite loose relative to what is done in practice. The best known provable bounds scale as $O(\sqrt{d}/\epsilon)$, where $d$ is the dimension of the space in which the diffusion is taking place (e.g. $d = 16384$ for Stable Diffusion) (Chen et al., 2023b), and $\epsilon$ is the target error. Even ignoring the dependence on $\epsilon$ and the hidden constant factor, this is at least $2 - 3\times$ larger than the default value of $50$ inference steps in Stable Diffusion.

In this work we consider a new approach for driving down the amount of compute that is provably needed to sample with diffusion models. Our approach is rooted in the *randomized midpoint method*, originally introduced by Shen and Lee (Shen & Lee, 2019) in the context of Langevin Monte Carlo for log-concave sampling. At a high level, this is a method for numerically solving differential equations where within every discrete window of time, one forms an unbiased estimate for the drift by evaluating it at a random "midpoint" (see Section 2.2 for a formal treatment). For sampling from log-concave densities, the number of iterations needed by their method scales with $d^{1/3}$, and this remains the best known bound in the "low-accuracy" regime.

While this method is well-studied in the log-concave setting (He et al., 2020; Yu et al., 2023; Yu & Dalalyana, 2024; Shen & Lee, 2019), its applicability to diffusion models has been unexplored both theoretically and empirically. Our first result uses the randomized midpoint method to obtain an improvement over the prior best known bound of $O(\sqrt{d}/\epsilon)$ for sampling arbitrary smooth distributions with diffusion models:

**Theorem 1.1** (Informal, see Theorem B.10). *Suppose that the data distribution $q$ has bounded second moment, its score functions $\nabla \ln q_t$ along the forward process are L-Lipschitz, and we are given score estimates which are L-Lipschitz and $\widetilde{O}(\frac{\epsilon}{d^{1/12}\sqrt{L}})^1$-close to $\nabla \ln q_t$ for all $t$. Then there is a diffusion-based sampler using these score estimates (see Algorithm 1) which outputs a sample whose law is $\epsilon$-close in total variation distance to $q$ using $\widetilde{O}(L^{5/3}d^{5/12}/\epsilon)$ iterations.*

Our algorithm is based on the ODE-based predictor-corrector algorithm introduced in (Chen et al., 2023b), but in place of the standard exponential integrator discretization in the predictor step, we employ randomized midpoint discretization. We note that in the domain of log-concave sampling, the result of Shen and Lee only achieves recovery in *Wasserstein distance*. Prior to our work, it was actually open whether one can achieve the same dimension dependence in total variation or KL divergence, for which the best known bound was $\widetilde{O}(\sqrt{d})$ (Ma et al., 2021; Zhang et al., 2023; Altschuler & Chewi, 2023). In contrast, our result circumvents this barrier by carefully trading off time spent in the corrector phase of the algorithm for time spent in the predictor phase. We defer the details of this, as well as other important technical hurdles, to Section 3.1.

Next, we turn to a different computational model: instead of quantifying the cost of an algorithm in terms of the total number of iterations, we consider the *parallel* setting where one has access to multiple processors and wishes to minimize the total number of parallel rounds needed to generate a single sample. This perspective has been explored in a recent empirical work (Shih et al., 2024), but to our knowledge, no provable guarantees were known for parallel sampling with diffusion models (see Section 1.1 for discussion of concurrent and independent work). Our second result provides the first such guarantee:

**Theorem 1.2** (Informal, see Theorem C.13). *Under the same assumptions on $q$ as in Theorem 1.1, and assuming that we are given score estimates which are $\widetilde{O}(\frac{\epsilon}{\sqrt{L}})$-close to $\nabla \ln q_t$ for all $t$, there is a diffusion-based sampler using these score estimates (see Algorithm 9) which outputs a sample whose law is $\epsilon$-close in total variation distance to $q$ using $\widetilde{O}(L \cdot \text{polylog}(Ld/\epsilon))$ parallel rounds.*

This result follows in the wake of several recent theoretical works on parallel sampling of log-concave densities using Langevin Monte Carlo (Anari et al., 2023; 2024; Shen & Lee, 2019). A common thread among these works is the observation that differential equations can be numerically solved via fixed point iteration (see Section 2.3 for details), and we adopt a similar perspective in the context of diffusions. To our knowledge this is the first provable guarantee for parallel sampling beyond the log-concave setting.

Finally, we show that, as a byproduct of our methods, we can actually obtain a similar dimension dependence of $\widetilde{O}(d^{5/12})$ as in Theorem 1.1 for log-concave sampling in TV, superseding the previously best known bound of $\widetilde{O}(\sqrt{d})$ mentioned above.

**Theorem 1.3** (Informal, see Theorem D.3). *Suppose distribution $q$ is $m$-strongly-log-concave, and its score function $\nabla \ln q$ is L-Lipschitz. Then, there is a underdamped-Langevin-based sampler that uses this score (Algorithm 11) and outputs a sample whose law is $\epsilon$-close in total variation to $q$ using $\widetilde{O}\left(d^{5/12}\left(\frac{L^{4/3}}{\epsilon^{2/3}m^{4/3}} + \frac{1}{\epsilon}\right)\right)$ iterations.*

## 1.1 RELATED WORK

Our discretization scheme is based on the randomized midpoint method of (Shen & Lee, 2019), which has been studied at length in the domain of log-concave sampling (He et al., 2020; Yu et al., 2023; Yu & Dalalyana, 2024).

---

[1] $\widetilde{O}(\cdot)$ hides polylogarithmic factors in $d, L, \epsilon$ and $\mathbb{E}_{x \sim q}[\|x\|^2]$

The proof of our parallel sampling result builds on the ideas of (Shen & Lee, 2019; Anari et al., 2023; 2024) on parallelizing the collocation method. These prior results were focused on Langevin Monte Carlo, rather than diffusion-based sampling. We review these ideas in Section 2.3.

In (Chen et al., 2023b), the authors proposed the predictor-corrector framework that we also use for analysing convergence guarantee of the probability flow ODE and which achieved iteration complexity scaling with $\widetilde{O}(\sqrt{d})$. In addition to this, there have been many works in recent years giving general convergence guarantees for diffusion models (De Bortoli et al., 2021; Block et al., 2022; De Bortoli, 2022; Lee et al., 2022; Liu et al., 2022; Pidstrigach, 2022; Wibisono & Yang, 2022; Chen et al., 2023c;d; Lee et al., 2023; Li et al., 2023; Benton et al., 2023; Chen et al., 2023b; Benton et al., 2024; Chen et al., 2023a; Gupta et al., 2023b). Of these, one line of work (Chen et al., 2023c; Lee et al., 2023; Chen et al., 2023a; Benton et al., 2024) analyzed DDPM, the stochastic analogue of the probability flow ODE, and showed $\tilde{O}(d)$ iteration complexity bounds. Another set of works (Chen et al., 2023b;d; Li et al., 2023; 2024) studied the probability flow ODE, for which our work provides a new discretization scheme for the probability flow ODE, that achieves a state-of-the-art $\widetilde{O}(d^{5/12})$ dimension dependence for sampling from a diffusion model.

**Concurrent work.** Here we discuss the independent works of (Chen et al., 2024) and (Kandasamy & Nagaraj, 2024). (Chen et al., 2024) gave an analysis for parallel sampling with diffusion models that also achieves a $\mathrm{polylog}(d)$ number of parallel rounds like in the present work. (Kandasamy & Nagaraj, 2024) showed an improved dimension dependence of $\widetilde{O}(d^{5/12})$ for log-concave sampling in total variation, similar to our analogous result, but via a different proof technique. In addition to this, they show a similar result when the distribution only satisfies a log-Sobolev inequality. They also show empirical results for diffusion models, showing that an algorithm inspired by the randomized midpoint method outperforms ODE based methods with similar compute. While their work builds on the randomized midpoint method, they do not theoretically analyze the diffusion setting and do not study parallel sampling.

## 2 PRELIMINARIES

### 2.1 PROBABILITY FLOW ODE

In this section we review basics about deterministic diffusion-based samplers; we refer the reader to (Chen et al., 2023b) for a more thorough exposition.

Let $q^*$ denote the data distribution over $\mathbb{R}^d$. We consider the standard Ornstein-Uhlenbeck (OU) *forward process*, i.e. the "VP SDE," given by

$$\mathrm{d}\overrightarrow{x_t} = -\overrightarrow{x_t}\,\mathrm{d}t + \sqrt{2}\,\mathrm{d}B_t \qquad \overrightarrow{x_0} \sim q^*, \tag{1}$$

where $(B_t)_{t\geq 0}$ denotes a standard Brownian motion in $\mathbb{R}^d$. This process converges exponentially quickly to its stationary distribution, the Gaussian distribution $\mathcal{N}(0, \mathrm{Id})$.

Suppose the OU process is run until terminal time $T > 0$, and for any $t \in [0, T]$, let $q_t^* \triangleq \mathrm{law}(\overrightarrow{x_t})$, i.e. the law of the forward process at time $t$. We will consider the *reverse process* given by the *probability flow ODE*

$$\mathrm{d}x_t = (x_t + \nabla \ln q_{T-t}(x_t))\,\mathrm{d}t. \tag{2}$$

This is a time-reversal of the forward process, so that if $x_0 \sim q_T$, then $\mathrm{law}(x_t) = q_{T-t}^*$. In practice, one initializes at $x_0 \sim \mathcal{N}(0, \mathrm{Id})$, and instead of using the exact *score function* $\nabla \ln q_{T-t}$, one uses estimates $\widehat{s}_{T-t} \approx \nabla \ln q_{T-t}$ which are learned from data. Additionally, the ODE is solved numerically using any of a number of discretization schemes. The theoretical literature on diffusion models has focused primarily on *exponential integration*, which we review next before turning to the discretization scheme, the *randomized midpoint method* used in the present work.

### 2.2 DISCRETIZATION SCHEMES

Suppose we wish to discretize the following semilinear ODE:

$$\mathrm{d}x_t = (x_t + f_t(x_t))\,\mathrm{d}t. \tag{3}$$

For our application we will eventually take $f_t \triangleq \widehat{s}_{T-t}$, but we use $f_t$ in this section to condense notation.

Suppose we want to discretize Equation (3) over a time window $[t_0, t_0 + h]$. The starting point is the integral formulation for this ODE:

$$x_{t_0+h} = e^h x_{t_0} + \int_{t_0}^{t_0+h} e^{t_0+h-t} f_t(x_t) \, dt \,. \tag{4}$$

Under the standard *exponential integrator* discretization, one would approximate the integrand by $e^{t_0+h-t} f_{t_0}(x_{t_0})$ and obtain the approximation

$$x_{t_0+h} \approx e^h x_{t_0} + (e^h - 1) f_{t_0}(x_{t_0}) \,. \tag{5}$$

The drawback of this discretization is that it uses an inherently *biased* estimate for the integral in Eq. equation 4. The key insight of (Shen & Lee, 2019) was to replace this with the following unbiased estimate

$$\int_{t_0}^{t_0+h} e^{t_0+h-t} f_t(x_t) \, dt \approx h e^{(1-\alpha)h} f_{t_0+\alpha h}(x_{t_0+\alpha h}) \,, \tag{6}$$

where $\alpha$ is a uniformly random sample from $[0, 1]$. While this alone does not suffice as the estimate depends on $x_{t_0+\alpha h}$, naturally we could iterate the above procedure again to obtain an approximation to $x_{t_0+\alpha h}$. It turns out though that even if we simply approximate $x_{t_0+\alpha h}$ using exponential integrator discretization, we can obtain nontrivial improvements in discretization error (e.g. our Theorem 1.1). In this case, the above sequence of approximations takes the following form:

$$x_{t_0+\alpha h} \approx e^{\alpha h} x_{t_0} + (e^{\alpha h} - 1) f_{t_0}(x_{t_0}) \tag{7}$$
$$x_{t_0+h} \approx e^h x_{t_0} + h e^{(1-\alpha)h} f_{t_0+\alpha h}(x_{t_0+\alpha h}) \,. \tag{8}$$

Note that a similar idea can be used to discretize stochastic differential equations, but in this work we only use it to discretize the probability flow ODE.

**Predictor-Corrector.** For important technical reasons, in our analysis we actually consider a slightly different algorithm than simply running the probability flow ODE with approximate score, Gaussian initialization, and randomized midpoint discretization. Specifically, we interleave the ODE with *corrector* steps that periodically inject noise into the sampling trajectory. We refer to the phases in which we are running the probability flow ODE as *predictor* steps.

The corrector step will be given by running *underdamped Langevin dynamics*. As our analysis of this will borrow black-box from bounds proven in (Chen et al., 2023b), we refer to Section B.2 for details.

## 2.3 PARALLEL SAMPLING

The scheme outlined in the previous section is a simple special case of the *collocation method*. In the context of the semilinar ODE from Eq. equation 3, the idea behind the collocation method is to solve the integral formulation of the ODE in Eq. equation 4 via fixed point iteration. For our parallel sampling guarantees, instead of choosing a single randomized midpoint $\alpha$, we break up the window $[t_0, t_0 + h]$ into $R$ sub-windows, select randomized midpoints $\alpha_1, \ldots, \alpha_R$ for these sub-windows, and approximate the trajectory of the ODE at any time $t_0 + i\delta$, where $\delta \triangleq h/R$, by

$$x_{t_0+\alpha_i h} \approx e^{\alpha_i h} x_{t_0} + \sum_{j=1}^{i} \left( e^{\alpha_i h - (j-1)\delta} - \max(e^{\alpha_i h - j\delta}, 1) \right) \cdot f_{t_0+\alpha_j h}(x_{t_0+\alpha_j h}) \,. \tag{9}$$

One can show that as $R \to \infty$, this approximation tends to an equality. For sufficiently large $R$, Eq. equation 9 naturally suggests a fixed point iteration that can be used to approximate each $x_{t_0+\alpha_i h}$, i.e. we can maintain a sequence of estimates $\widehat{x}_{t_0+\alpha_i h}^{(k)}$ defined by the iteration

$$\widehat{x}_{t_0+\alpha_i h}^{(k)} \leftarrow e^{\alpha_i h} \widehat{x}_{t_0}^{(k-1)} + \sum_{j=1}^{i} \left( e^{\alpha_i h - (j-1)\delta} - \max(e^{\alpha_i h - j\delta}, 1) \right) \cdot f_{t_0+\alpha_j h}(\widehat{x}_{t_0+\alpha_j h}^{(k-1)}) \,, \tag{10}$$

for $k$ ranging from 1 up to some sufficiently large $K$. Finally, analogously to Eq. equation 8, we can estimate $x_{t_0+h}$ via

$$x_{t_0+h} \approx e^h \widehat{x}_{t_0}^{(K)} + \delta \sum_{i=1}^{R} e^{(1-\alpha_i)h} f_{t_0+\alpha_i h}(\widehat{x}_{t_0+\alpha_i h}^{(K)}). \qquad (11)$$

The key observation, made in (Shen & Lee, 2019) and also in related works of (Anari et al., 2024; Shih et al., 2024; Anari et al., 2023), is that for any fixed round $k$, all of the iterations Eq. equation 10 for different choices of $i = 1, \ldots, R$ can be computed in parallel. With $R$ parallel processors, one can thus compute the estimate for $x_{t_0+h}$ in $K$ parallel rounds, with $O(KR)$ total work.

## 2.4 ASSUMPTIONS

Throughout the paper, for our diffusion results, we will make the following standard assumptions on the data distribution and score estimates.

**Assumption 2.1** (Bounded Second Moment)**.**

$$\mathfrak{m}_2^2 := \mathop{\mathbb{E}}_{x \sim q_0} \left[ \|x\|^2 \right] < \infty.$$

**Assumption 2.2** (Lipschitz Score)**.** *For all $t$, the score $\nabla \ln q_t$ is $L$-Lipschitz.*

**Assumption 2.3** (Lipschitz Score estimates)**.** *For all $t$ for which we need to estimate the score function in our algorithms, the score estimate $\widehat{s}_t$ is $L$-lipschitz.*

**Assumption 2.4** (Score Estimation Error)**.** *For all $t$ for which we need to estimate the score function in our algorithms,*

$$\mathop{\mathbb{E}}_{x_t \sim q_t} \left[ \|\widehat{s}_t(x_t) - \nabla \ln q_t(x_t)\|^2 \right] \leq \epsilon_{\mathrm{sc}}^2.$$

## 3 TECHNICAL OVERVIEW

Here we provide an overview of our sequential and parallel algorithms, along with the analysis of our iteration complexity bounds. We begin with a description of the sequential algorithm.

### 3.1 SEQUENTIAL ALGORITHM

Following the framework of (Chen et al., 2023b), our algorithm consists of "predictor" steps interspersed with "corrector" steps, with the time spent on each carefully tuned to obtain our final $\widetilde{O}(d^{5/12})$ dimension dependence. We first describe our predictor step – this is the piece of our algorithm that makes use of the Shen and Lee's randomized midpoint method (Shen & Lee, 2019).

---

**Algorithm 1** PREDICTORSTEP (SEQUENTIAL)

---

**Input parameters:**

- Starting sample $\widehat{x}_0$, Starting time $t_0$, Number of steps $N$, Step sizes $h_{n \in [0,\ldots,N-1]}$, Score estimates $\widehat{s}_t$

1. For $n = 0, \ldots, N-1$:

    (a) Let $t_n = t_0 - \sum_{i=0}^{n-1} h_i$
    (b) Randomly sample $\alpha$ uniformly from $[0, 1]$.
    (c) Let $\widehat{x}_{n+\frac{1}{2}} = e^{\alpha h_n} \widehat{x}_n + \left( e^{\alpha h_n} - 1 \right) \widehat{s}_{t_n}(\widehat{x}_n) ds$
    (d) Let $\widehat{x}_{n+1} = e^{h_n} \widehat{x}_n + h_n \cdot e^{(1-\alpha)h_n} \widehat{s}_{t_n - \alpha h_n}(\widehat{x}_{n+\frac{1}{2}})$

2. Let $t_N = t_0 - \sum_{i=0}^{N-1} h_i$
3. Return $\widehat{x}_N, t_N$.

---

The main difference between the above and the predictor step of (Chen et al., 2023b) are steps $1(b)$ – $1(d)$. $1(b)$ and $1(c)$ together compute a randomized midpoint, and $1(d)$ uses this midpoint to obtain an approximate solution to the integral of the ODE. We describe these steps in more detail in Section 3.3.

Next, we describe the "corrector" step, introduced in (Chen et al., 2023b). First, recall the underdamped Langevin ODE:

$$
\begin{aligned}
\mathrm{d}\widehat{x}_t &= \widehat{v}_t \, \mathrm{d}t \\
\mathrm{d}\widehat{v}_t &= \left( \widehat{s}(\widehat{x}_{\lfloor t/h \rfloor h}) - \gamma \widehat{v}_t \right) \mathrm{d}t + \sqrt{2\gamma} \, \mathrm{d}B_t
\end{aligned}
\tag{12}
$$

Here $\widehat{s}$ is our $L^2$ accurate score estimate for a fixed time (say $t$). Then, the corrector step is described below.

---

**Algorithm 2** CORRECTORSTEP (SEQUENTIAL)

---

**Input parameters:**

- Starting sample $\widehat{x}_0$, Total time $T_{\mathrm{corr}}$, Step size $h_{\mathrm{corr}}$, Score estimate $\widehat{s}$

1. Run underdamped Langevin Monte Carlo in equation 12 for total time $T_{\mathrm{corr}}$ using step size $h_{\mathrm{corr}}$, and let the result be $\widehat{x}_N$.
2. Return $\widehat{x}_N$.

---

Finally, Algorithm 3 below puts the predictor and corrector steps together to give our final sequential algorithm.

---

**Algorithm 3** SEQUENTIALALGORITHM

---

**Input parameters:**

- Start time $T$, End time $\delta$, Corrector steps time $T_{\mathrm{corr}} \lesssim 1/\sqrt{L}$, Number of predictor-corrector steps $N_0$, Predictor step size $h_{\mathrm{pred}}$, Corrector step size $h_{\mathrm{corr}}$, Score estimates $\widehat{s}_t$

1. Draw $\widehat{x}_0 \sim \mathcal{N}(0, I_d)$.
2. For $n = 0, \ldots, N_0 - 1$:
    - (a) Starting from $\widehat{x}_n$, run Algorithm 1 with starting time $T - n/L$ using step sizes $h_{\mathrm{pred}}$ for all $N$ steps, with $N = \frac{1}{Lh_{\mathrm{pred}}}$, so that the total time is $1/L$. Let the result be $\widehat{x}'_{n+1}$.
    - (b) Starting from $\widehat{x}'_{n+1}$, run Algorithm 1 for total time $T_{\mathrm{corr}}$ with step size $h_{\mathrm{corr}}$ and score estimate $\widehat{s}_{T-(n+1)/L}$ to obtain $\widehat{x}_{n+1}$.
3. Starting from $\widehat{x}_{N_0}$, run Algorithm 4 with starting time $T - N_0/L$ using step sizes $h_{\mathrm{pred}}/2, h_{\mathrm{pred}}/4, h_{\mathrm{pred}}/8, \ldots, \delta$ to obtain $\widehat{x}'_{N_0+1}$.
4. Starting from $\widehat{x}'_{N_0+1}$, run Algorithm 2 for total time $T_{\mathrm{corr}}$ with step size $h_{\mathrm{corr}}$ and score estimate $\widehat{s}_\delta$ to obtain $\widehat{x}_{N_0+1}$.
5. Return $\widehat{x}_{N_0+1}$.

---

For the final setting of parameters in Algorithm 3, see Theorem B.10. Now, we describe the analysis of the above algorithm in detail.

### 3.2 PREDICTOR-CORRECTOR FRAMEWORK

The general framework of our algorithm closely follows that of (Chen et al., 2023b), which proposed to run the (discretized) reverse ODE but interspersed with "corrector" steps given by running underdamped Langevin dynamics. The idea is that the "predictor" steps where the discretized reverse

ODE is being run keep the sampler close to the true reverse process in Wasserstein distance, but they cannot be run for too long before potentially incurring exponential blowups. The main purpose of the corrector steps is then to inject stochasticity into the trajectory of the sampler in order to convert closeness in Wasserstein to closeness in KL divergence. This effectively allows one to "restart the coupling" used to control the predictor steps. For technical reasons that are inherited from (Chen et al., 2023b), for most of the reverse process the predictor steps (Step 2(a)) are run with a fixed step size, but at the end of the reverse process (Step 3), they are run with exponentially decaying step sizes.

We follow the same framework, and the core of our result lies in refining the algorithm and analysis for the predictor steps by using the randomized midpoint method. Below, we highlight our key technical steps.

### 3.3 PREDICTOR STEP – IMPROVED DISCRETIZATION ERROR WITH RANDOMIZED MIDPOINTS

Here, we explain the main idea behind why randomized midpoint allows us to achieve improved dimension dependence. We first focus on the analysis of the predictor (Algorithm 1) and restrict our attention to running the reverse process for a small amount of time $h \ll 1/L$.

We begin by recalling the dimension dependence achieved by the standard exponential integrator scheme. One can show (see e.g. Lemma 4 in (Chen et al., 2023b)) that if the true reverse process and the discretized reverse process are both run for small time $h$ starting from the same initialization, the two processes drift by a distance of $O(d^{1/2}h^2)$. By iterating this coupling $O(1/h)$ times, we conclude that in an $O(1)$ window of time, the processes drift by a distance of $O(d^{1/2}h)$. To ensure this is not too large, one would take the step size $h$ to be $O(1/\sqrt{d})$, thus obtaining an iteration complexity of $O(\sqrt{d})$ as in (Chen et al., 2023b).

The starting point in the analysis of randomized midpoint is to instead track the *squared* displacement between the two processes instead. Given two neighboring time steps $t - h$ and $t$ in the algorithm, let $x_t$ denote the true reverse process at time $t$, and let $\widehat{x}_t$ denote the algorithm at time $t$ (in the notation of Algorithm 2, this is $\widehat{x}_n$ for some $n$, but we use $t$ in the discussion here to make the comparison to the true reverse process clearer). Note that $\widehat{x}_t$ depends on the choice of randomized midpoint $\alpha$ (see Step 1(b)). One can bound the squared displacement $\mathbb{E} \|x_t - \widehat{x}_t\|^2$ as follows. Let $y_t$ be the result of running the reverse process for time $h$ starting from $\widehat{x}_{t-h}$. Then by writing $x_t - \widehat{x}_t$ as $(x_t - y_t) - (\widehat{x}_t - y_t)$ and applying Young's inequality, we obtain

$$\underset{\widehat{x}_{t-h}, \alpha}{\mathbb{E}} \|x_t - \widehat{x}_t\|^2 \leq \left(1 + \frac{Lh}{2}\right) \underset{\widehat{x}_{t-h}}{\mathbb{E}} \|x_t - y_t\|^2 + \frac{2}{Lh} \underset{\widehat{x}_{t-h}}{\mathbb{E}} \|\underset{\alpha}{\mathbb{E}} \widehat{x}_t - y_t\|^2 + \underset{\widehat{x}_{t-h}}{\mathbb{E}} \underset{\alpha}{\mathbb{E}} \|\widehat{x}_t - y_t\|^2 . \quad (13)$$

For the first term, because $x_t$ and $y_t$ are the result of running the same ODE on initializations $x_{t-h}$ and $\widehat{x}_{t-h}$, the first term is close to $\mathbb{E} \|x_{t-h} - \widehat{x}_{t-h}\|^2$ provided $h \ll 1/L$. The upshot is that the squared displacement at time $t$ is at most the squared displacement at time $t - h$ plus the remaining two terms on the right of Equation (13).

The main part of the proof lies in bounding these two terms, which can be thought of as "bias" and "variance" terms respectively. The variance term can be shown to scale with the square of the aforementioned $O(d^{1/2}h^2)$ displacement bound that arises in the exponential integrator analysis, giving $O(dh^4)$:

**Lemma 3.1** (Informal, see Lemma B.4 for formal statement). *If $h \lesssim \frac{1}{L}$ and $T - t \geq (T - t - h)/2$, then*

$$\underset{\widehat{x}_{t-h}}{\mathbb{E}} \underset{\alpha}{\mathbb{E}} \|\widehat{x}_t - y_t\|^2 \lesssim L^2 dh^4 \left(L \vee \frac{1}{T - (t - h)}\right) + h^2 \epsilon_{\text{sc}}^2 + L^2 h^2 \underset{\widehat{x}_{t-h}}{\mathbb{E}} \|x_{t-h} - \widehat{x}_{t-h}\|^2 .$$

Note that in this bound, in addition to the $O(dh^4)$ term and a term for the score estimation error, there is an additional term which depends on the squared displacement from the previous time step. Because the prefactor $L^2 h^2$ is sufficiently small, this will ultimately be negligible.

The upshot of the above Lemma is that if the bias term is of lower order, then this means that the squared displacement essentially increases by $O(dh^4)$ with every time step of length $h$. Over $O(1/h)$ such steps, the total squared displacement is $O(dh^3)$, so if we take the step size $h$ to be $O(1/d^{1/3})$, this suggests an improved iteration complexity of $O(d^{1/3})$.

Arguing that the bias term $\frac{2}{Lh}\mathbb{E}_{\widehat{x}_{t-h}}\|\mathbb{E}_\alpha\widehat{x}_t - y_t\|^2$ is dominated by the variance term is where it is crucial that we use randomized midpoint instead of exponential integrator. But recall that the randomized midpoint method was engineered so that it would give an unbiased estimate for the true solution to the reverse ODE if the estimate of the trajectory at the randomized midpoint were exact. In reality we only have an approximation to the latter, but as we show, the error incurred by this is indeed of lower order (see Lemma B.3). One technical complication that arises here is that the relevant quantity to bound is the distance between the true process at the randomized midpoint versus the algorithm, when both are initialized at an intermediate point in the algorithm's trajectory. Bounding such quantities in expectation over the randomness of the algorithm's trajectory can be difficult, but our proof identifies a way of "offloading" some of this difficulty by absorbing some excess terms into a term of the form $\|x_{t-h} - \widehat{x}_{t-h}\|^2$, i.e. the squared displacement from the previous time step. Concretely, we obtain the following bound on the bias term:

**Lemma 3.2** (Informal, see Lemma B.2 for formal statement)**.**

$$\mathop{\mathbb{E}}_{\widehat{x}_{t-h}}\|\mathbb{E}_\alpha\widehat{x}_t - y_t\|^2 \lesssim L^4 dh^6\left(L \vee \frac{1}{T-t+h}\right) + h^2\epsilon_{\text{sc}}^2 + L^2h^2 E_{\widehat{x}_{t-h}}\|x_{t-h} - \widehat{x}_{t-h}\|^2$$

### 3.4 SHORTENING THE CORRECTOR STEPS

While we have outlined how to improve the predictor step in the framewok of (Chen et al., 2023b), it is quite unclear whether the same can be achieved for the corrector step. Whereas the the former is geared towards closeness in Wasserstein distance, the latter is geared towards closeness in KL divergence, and it is a well-known open question in the log-concave sampling literature to obtain analogous discretization bounds in KL for the randomized midpoint method (Chewi, 2023).

We will sidestep this issue and argue that even using exponential integrator discretization of the underdamped Langevin dynamics will suffice for our purposes, by simply shortening the amount of time for which each corrector step is run.

First, let us briefly recall what was shown in (Chen et al., 2023b) for the corrector step. If one runs underdamped Langevin dynamics with stationary distribution $q$ for time $T$ and exponential integrator discretization with step size $h$ starting from two distributions $p$ and $q$, then the resulting distributions $p'$ and $q$ satisfy

$$\mathsf{TV}(p', q) \lesssim \frac{W_2(p, q)}{L^{1/4}T^{3/2}} + L^{3/4}T^{1/2}d^{1/2}h, \tag{14}$$

where $L$ is the Lipschitzness of $\nabla\ln q$ (see Theorem B.6). At first glance this appears insufficient for our purposes: because of the $d^{1/2}h$ term coming from the discretization error, we would need to take step size $h = 1/\sqrt{d}$, which would suggest that the number of iterations must scale with $\sqrt{d}$.

To improve the dimension dependence for our overall predictor-corrector algorithm, we observe that if we take $T$ itself to be smaller, then we can take $h$ to be larger while keeping the discretization error in Equation (14) sufficiently small. Of course, this comes at a cost, as $T$ also appears in the term $\frac{W_2(p,q)}{L^{1/4}T^{3/2}}$ in Equation (14). But in our overall proof, the $W_2(p, q)$ term is bounded by the predictor analysis. There, we had quite a bit of slack: even with step size as large as $1/d^{1/3}$, we could achieve small Wasserstein error. By balancing appropriately, we get our improved dimension dependence.

### 3.5 PARALLEL ALGORITHM

Now, we summarize the main proof ideas for our parallel sampling result. In Section 2.3, we described how to approximately solve the reverse ODE over time $h$ by running $K$ rounds of the iteration in Equation (10). In our final algorithm, we will take $h$ to be *dimension-independent*, namely $h = \Theta(1/\sqrt{L})$, so that the main part of the proof is to bound the discretization error incurred over each of these time windows of length $h$. As in the sequential analysis, we will interleave these "predictor" steps with corrector steps given by (parallelized) underdamped Langevin dynamics.

We begin by describing the parallel predictor step. Suppose we have produced an estimate for the reverse process at $t_0$ and now wish to solve the ODE from time $t_0$ to $t_0 + h$. We initialize at $\{\widehat{x}_{t_0+\alpha_i h}^{(0)}\}_{i\in[R]}$ via exponential integrator steps starting from the beginning of the window – see Line

1(c) in Algorithm 9 (this can be thought of as the analogue of Equation (7) used in the sequential algorithm). The key difference relative to the sequential algorithm is that here, because the length of the window is dimension-free, the discretization error incurred by this initialization is too large and must be refined using the fixed point iteration in Equation (10). The main step is then to show that with each iteration of Equation (10), the distance to the true reverse process contracts:

**Lemma 3.3** (Informal, see Lemma C.2 for formal statement). *Suppose $h \lesssim 1/L$. If $y_t$ denotes the solution of the true ODE starting at $\widehat{x}_{t_0}$ and running until time $t_0 + \alpha_i h$, then for all $k \in \{1, \cdots K\}$ and $i \in \{1, \cdots, R\}$,*

$$
\underset{\widehat{x}_{t_0}, \alpha_1, \cdots \alpha_R}{\mathbb{E}} \left\| \widehat{x}^{(k)}_{t_0 + \alpha_i h} - y_{t_0 + \alpha_i h} \right\|^2 \lesssim \left( 8h^2 L^2 \right)^k \cdot \left( \frac{1}{R} \sum_{j=1}^{R} \underset{\widehat{x}_{t_0}, \alpha_j}{\mathbb{E}} \left\| \widehat{x}^{(0)}_{t_0 + \alpha_j h} - y_{t_0 + \alpha_j h} \right\|^2 \right)
$$
$$
+ h^2 \left( \epsilon_{\mathrm{sc}}^2 + \frac{L^2 d h^2}{R^2} (L \vee \frac{1}{T - t_0 + h}) + L^2 \cdot \underset{\widehat{x}_{t_0}}{\mathbb{E}} \left\| \widehat{x}_{t_0} - x_{t_0} \right\|^2 \right) ,
$$
(15)

*where $\widehat{x}_{t_0}$ is the iterate of the algorithm from the previous time window, and $x_{t_0}$ is the corresponding iterate in the true ODE.*

In particular, because $h$ is at most a small multiple of $1/L$, the prefactor $(8h^2 L^2)^k$ is exponentially decaying in $k$, so that the error incurred by the estimate $\widehat{x}^{(k)}_{t_0 + \alpha_i h}$ is contracting with each fixed point iteration. Because the initialization is at distance $\mathrm{poly}(d)$ from the true process, $O(\log d)$ rounds of contraction thus suffice, which translates to $O(\log d)$ parallel rounds for the sampler. The rest of the analysis of the predictor step is quite similar to the analogous proofs for the sequential algorithm (i.e. Lemma C.4 and Lemma C.5 give the corresponding bias and variance bounds).

One shortcoming of the predictor analysis is that the contraction achieved by fixed point iteration ultimately bottoms out at error which scales with $d/R^2$ (see the second term in Equation (15)). In order for the discretization error to be sufficiently small, we thus have to take $R$, and thus the total work of the algorithm, to scale with $O(\sqrt{d})$. So in this case we do not improve over the dimension dependence of (Chen et al., 2023b), and instead the improvement is in obtaining a parallel algorithm.

For the corrector analysis, we mostly draw upon the recent work of (Anari et al., 2024) which analyzed a parallel implementation of the underdamped Langevin dynamics. While their guarantee focuses on sampling from log-concave distributions, implicit in their analysis is a bound for general smooth distributions on how much the law of the algorithm and the law of the true process drift apart in a bounded time window (see Lemma C.8). This bound suffices for our analysis of the corrector step, and we can conclude the following:

**Theorem 3.4** (Informal, see Theorem C.12). *Let $\beta \geq 1$ be an adjustable parameter. Let $p'$ denote the law of the output of running the parallel corrector (see Algorithm 8) for total time $1/\sqrt{L}$ and step size $h$, using an $\epsilon_{\mathrm{sc}}$-approximate estimate for $\nabla \ln q$ and starting from a sample from another distribution $p$.*

$$
\mathsf{TV}(p', q) \lesssim \sqrt{\mathsf{KL}(p', q)} \lesssim \frac{\epsilon_{\mathrm{sc}}}{\sqrt{L}} + \frac{\epsilon}{\beta} + \frac{\epsilon}{\beta \sqrt{d}} \cdot W_2(p, q) .
$$

*Furthermore, this algorithm uses $\widetilde{\Theta}(\beta \sqrt{d}/\epsilon)$ score evaluations over $\Theta(\log(\beta^2 d/\epsilon^2))$ parallel rounds.*

Overall, parallel algorithm is somewhat different from the parallel sampler developed in the empirical work of (Shih et al., 2024), even apart from the fact that we use randomized midpoint discretization and corrector steps. The reason is that our algorithm applies collocation to fixed windows of time, whereas the algorithm of (Shih et al., 2024) utilizes a sliding window approach that proactively shifts the window forward as soon as the iterates at the start of the previous window begin to converge. We leave rigorously analyzing the benefits of this approach as an interesting future direction.

## 3.6 LOG-CONCAVE SAMPLING IN TOTAL VARIATION

Finally, we briefly summarize the simple proof for our result on log-concave sampling in TV, which achieves the best known dimension dependence of $\widetilde{O}(d^{5/12})$. Our main observation is that Shen

and Lee's randomized midpoint method (Shen & Lee, 2019) applied to the underdamped Langevin process gives a *Wasserstein* guarantee for log-concave sampling, while the corrector step of (Chen et al., 2023b) can convert a Wasserstein guarantee to closeness in TV. Thus, we can simply run the randomized midpoint method, followed by the corrector step to achieve closeness in TV. Carefully tuning the amount of time spend and step sizes for each phase of this algorithm yields our improved dimension dependence – see Appendix D for the full proof.

## 4  DISCUSSION AND FUTURE WORK

In this work, we showed that it is possible to leverage Shen and Lee's randomized midpoint method (Shen & Lee, 2019) to achieve the best known dimension dependence for sampling from arbitrary smooth distributions in TV using diffusion. We also showed how to parallelize our algorithm, and showed that $\widetilde{O}(\log^2 d)$ parallel rounds suffice for sampling. These constitute the first provable guarantees for parallel sampling with diffusion models. Finally, we showed that our techniques can be used to obtain an improved dimension dependence for log-concave sampling in TV.

We note that relative to (Chen et al., 2023b), our result requires a slightly stronger guarantee on the score estimation error, by a $d^{1/12}$ factor; we believe this is an artifact of our analysis, and it would be interesting to remove this dependence in future work. Importantly however, it was not known how to achieve an improvement over the $O(\sqrt{d})$ dependence shown in that paper even in case that the scores are known *exactly* prior to the present work. Moreover, another line of work (Li et al., 2023; 2024; Dou et al., 2024) analyzing diffusion sampling makes the *stronger* assumption that the score estimation error is $\widetilde{O}\left(\frac{\epsilon}{\sqrt{d}}\right)$, an assumption stronger than ours by a $d^{5/12}$ factor; this does not detract from the importance of these works, and we feel the same is true in our case.

We also note that our diffusion results require smoothness assumptions – we assume that the true score, as well as our score estimates are $L$-Lipschitz. Although this assumption is standard in the literature, recent work (Chen et al., 2023a; Benton et al., 2024) has analyzed DDPM in the absence of these assumptions, culminating in a $\widetilde{O}(d)$ dependence for sampling using a discretization of the reverse SDE. However, unlike in the smooth case, it is not known whether even a *sublinear* in $d$ dependence is possible without smoothness assumptions via any algorithm. We leave this as an interesting open question for future work.

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

As a notational remark, in the proofs to follow we will sometimes use the notation $\mathsf{KL}(x \parallel y)$, $W_2(x, y)$, and $\mathsf{TV}(x, y)$ for random variables $x$ and $y$ to denote the distance between their associated probability distributions. Also, throughout the Appendix, we use $t$ to denote time in the forward process.

## A  EXPERIMENTS

**Experimental setup.** For all of our experiments, we use one NVIDIA A100 GPU. We evaluate our (sequential) randomized midpoint scheduler (predictor step only), a deterministic midpoint scheduler (where the midpoint is the average of the start and end times of a step), the default DDIM scheduler, and the default DDPM scheduler on the following datasets: CIFAR-10 (generated image dimension: $32 \times 32$), and CelebAHQ (generated image dimension: $256 \times 256$). To obtain score estimations for both of the schedulers, we use public pretrained DDPM models release by (Ho et al., 2020) for the two datasets[2], and generate 50k sample images for each dataset.

**Evaluation.** The performance of our scheduler and the default DDIM scheduler is evaluated by comparing the Fréchet Inception Distance (FID) scores (Heusel et al., 2017), which measure the quality of generated samples relative to the target distribution. Specifically, we use pytorch-fid[3]. The most expensive part of a diffusion scheduling algorithm is the number of evaluations of the score estimation function, as it involves calling the pre-trained neural network. To obtain a fair comparison, we maintain the same number of function calls (NFE) between the schedulers. Since randomized and deterministic midpoint takes two NFE per step, while DDIM and DDPM takes one NFE per step, this means that randomized and deterministic midpoint schedulers takes twice the step size of DDIM and DDPM in our experiments. Our results can be found in Figure 1.

---

[2]These pretrained models can be found respectively at https://heibox.uni-heidelberg.de/f/2e4f01e2d9ee49bab1d5/?dl=1; https://huggingface.co/google/ddpm-ema-celebahq-256.

[3]https://pypi.org/project/pytorch-fid/

**Discussion.** In our experiments, randomized midpoint consistently outperforms DDIM and DDPM on FID scores, which matches our theoretical analysis. The randomized midpoint appears to perform better than the deterministic midpoint on the CIFAR-10 dataset, while showing comparable performance on the CelebAHQ dataset. We suspect that this relative performance difference is related to the empirical bias of the deterministic midpoint estimator.

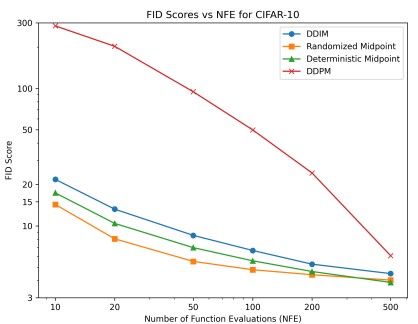 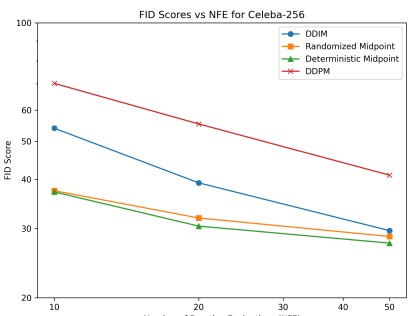

Figure 1: Number of Function Calls vs. FID score for CIFAR-10 and CelebAHQ Datasets (both axes are on a log scale).

## B  SEQUENTIAL ALGORITHM

In this section, we describe our sequential randomized-midpoint-based algorithm in detail. Following the framework of (Chen et al., 2023b), we begin by describing the *predictor Step* and show in Lemma B.5 that in $\widetilde{O}(d^{1/3})$ steps (ignoring other dependencies), when run for time $t$ at most $O(\frac{1}{L})$ starting from $t_n$, it produces a sample that is close to the true distribution at time $t_n - t$. Then, we show that the *corrector step* can be used to convert our $W_2$ error to error in TV distance by running the underdamped Langevin Monte Carlo algorithm, as described in (Chen et al., 2023b). We show in B.8 that if we run our predictor and corrector steps in succession for a careful choice of times, we obtain a sample that is close to the true distribution in TV using just $\widetilde{O}(d^{5/12})$ steps, but covering a time $O\left(\frac{1}{L}\right)$. Finally, in Theorem B.10, we iterate this bound $\widetilde{O}(\log^2 d)$ times to obtain our final iteration complexity of $\widetilde{O}(d^{5/12})$.

### B.1  PREDICTOR STEP

To show the $\widetilde{O}(d^{1/3})$ dependence on dimension for the predictor step, we will, roughly speaking, show that its bias after *one step* is bounded by $\approx O_L\left(dh^6\right)$ in Lemma B.3, and that the variance is bounded by $O_L\left(dh^4\right)$ in Lemma B.4. Then, iterating these bounds $\approx \frac{1}{h}$ times as shown in Lemma B.5 will give error $O_L\left(dh^4 + dh^3\right)$ in squared Wasserstein Distance.

**Lemma B.1** (Naive ODE Coupling). *Consider two variables $x_0, x_0'$ starting at time $t_0$, and consider the result of running the true ODE for time $h$, and let the results be $x_1, x_1'$. For $L \geq 1$, $h \leq 1/L$, we have*

$$\|x_1 - x_1'\|^2 \leq \exp(O(Lh))\|x_0 - x_0'\|^2$$

*Proof.* Recall that the true ODE is given by

$$dx_t = (x_t + \nabla \ln q_{T-t}(x_t))dt$$

So,

$$\partial_t \|x_t - x_t'\|^2 = 2\langle x_t - x_t', \partial_t x_t - \partial_t x_t'\rangle$$
$$= 2\langle x_t - x_t', x_t - x_t' + \nabla \ln q_{T-t}(x_t) - \nabla \ln q_{T-t}(x_t')\rangle$$
$$\lesssim L\|x_t - x_t'\|^2$$

---

**Algorithm 4** PREDICTORSTEP (SEQUENTIAL)

---

**Input parameters:**

- Starting sample $\widehat{x}_0$, Starting time $t_0$, Number of steps $N$, Step sizes $h_{n\in[0,\ldots,N-1]}$, Score estimates $\widehat{s}_t$

1. For $n = 0, \ldots, N-1$:

   (a) Let $t_n = t_0 - \sum_{i=0}^{n-1} h_i$

   (b) Randomly sample $\alpha$ uniformly from $[0, 1]$.

   (c) Let $\widehat{x}_{n+\frac{1}{2}} = e^{\alpha h_n}\widehat{x}_n + \left(e^{\alpha h_n} - 1\right)\widehat{s}_{t_n}(\widehat{x}_n)ds$

   (d) Let $\widehat{x}_{n+1} = e^{h_n}\widehat{x}_n + h_n \cdot e^{(1-\alpha)h_n}\widehat{s}_{t_n - \alpha h_n}(\widehat{x}_{n+\frac{1}{2}})$

2. Let $t_N = t_0 - \sum_{i=0}^{N-1} h_i$

3. Return $\widehat{x}_N, t_N$.

---

So,

$$\|x_1 - x_1'\|^2 \leq \exp\left(O(Lh)\right)\|x_0 - x_0'\|^2. \qquad \square$$

**Lemma B.2.** *Suppose $L \geq 1$. In Algorithm 4, for all $n \in \{0, \ldots, N-1\}$, let $x_n^*(t)$ be the solution of the true ODE starting at $\widehat{x}_n$ at time $t_n$, running until time $t_n - t$. If $h_n \lesssim \frac{1}{L}$ and $t_n - h_n \geq t_n/2$, we have*

$$\mathbb{E}\left\|h_n e^{(1-\alpha)h_n}\widehat{s}_{t_n - \alpha h_n}(\widehat{x}_{n+\frac{1}{2}}) - h_n e^{(1-\alpha)h_n}\nabla \ln q_{t_n - \alpha h_n}(x_n^*(\alpha h_n))\right\|^2$$

$$\lesssim h_n^2 \epsilon_{\text{sc}}^2 + L^4 dh_n^6\left(L \vee \frac{1}{t_n}\right) + L^2 h_n^2\,\mathbb{E}\left\|\widehat{x}_n - x_{t_n}\right\|^2,$$

*where $\mathbb{E}$ refers to the expectation over the initial choice $\widehat{x}_0 \sim q_{t_0}$.*

*Proof.* For the proof, we will let $h := h_n$. It suffices to show that

$$\mathbb{E}\left\|\widehat{s}_{t_n - \alpha h}(\widehat{x}_{n+\frac{1}{2}}) - \nabla \ln q_{t_n - \alpha h}(x_n^*(\alpha h))\right\|^2 \lesssim \epsilon_{\text{sc}}^2 + L^4 dh^4\left(L \vee \frac{1}{t_n}\right) + L^2\,\mathbb{E}\left\|\widehat{x}_n - x_{t_n}\right\|^2 \quad (16)$$

We can separate the above quantity into three parts: the expected difference in score estimation function $s(\cdot)$ evaluated at $\hat{x}_{n+1/2}$ and $x_{t_n - \alpha h}$; the expected difference in the true score function $q(\cdot)$ evaluated at $x_n^*(\alpha h)$ and $x_{t_n - \alpha h}$; and the expected difference between $s(\cdot)$ and $q(\cdot)$ both evaluated at $x_{t_n - \alpha h}$. By our Lipschitz assumption on both the score and score estimation function (Theorem 2.2 and Theorem 2.3), the first two terms can be bounded by $L^2$ times the difference in estimated and true sample $x$. By Theorem 2.4, the last term can be bounded by $\epsilon_{sc}^2$. Formally, we have

$$\mathbb{E}\left\|\widehat{s}_{t_n - \alpha h}(\widehat{x}_{n+\frac{1}{2}}) - \nabla \ln q_{t_n - \alpha h}(x_n^*(\alpha h))\right\|^2$$

$$\lesssim \mathbb{E}\left\|\widehat{s}_{t_n - \alpha h}(\widehat{x}_{n+\frac{1}{2}}) - \widehat{s}_{t_n - \alpha h}(x_{t_n - \alpha h})\right\| + \mathbb{E}\left\|\nabla \ln q_{t_n - \alpha h}(x_n^*(\alpha h)) - \nabla \ln q_{t_n - \alpha h}(x_{t_n - \alpha h})\right\|^2$$

$$+ \mathbb{E}\left\|\widehat{s}_{t_n - \alpha h}(x_{t_n - \alpha h}) - \nabla \ln q_{t_n - \alpha h}(x_{t_n - \alpha h})\right\|^2$$

$$\lesssim L^2\,\mathbb{E}\left\|\widehat{x}_{n+\frac{1}{2}} - x_{t_n - \alpha h}\right\|^2 + L^2\,\mathbb{E}\left\|x_{t_n - \alpha h} - x_n^*(\alpha h)\right\|^2 + \epsilon_{\text{sc}}^2$$

$$\lesssim L^2\,\mathbb{E}\left\|\widehat{x}_{n+\frac{1}{2}} - x_n^*(\alpha h)\right\|^2 + L^2\,\mathbb{E}\left\|x_n^*(\alpha h) - x_{t_n - \alpha h}\right\|^2 + \epsilon_{\text{sc}}^2. \quad (17)$$

By Theorem B.1, for any $s \in [0, h]$,

$$\|x_n^*(s) - x_{t_n - s}\|^2 \lesssim \exp\left(O(Lh)\right)\|\widehat{x}_n - x_{t_n}\|^2 \lesssim \|\widehat{x}_n - x_{t_n}\|^2, \quad (18)$$

thus the second term in Equation (17) is bounded by $L^2\,\mathbb{E}\left\|x_n^*(\alpha h) - x_{t_n - \alpha h}\right\|^2 \lesssim L^2\|\widehat{x}_n - x_{t_n}\|^2$. Now, we just need to bound $\mathbb{E}\left\|\widehat{x}_{n+\frac{1}{2}} - x_n^*(\alpha h)\right\|^2$.

Note that $x_n^*(\alpha h)$ is the solution to the following ODE run for time $\alpha h$, starting at $\widehat{x}_n$ at time $t_n$:

$$dx_t = (x_t + \nabla \ln q_t(x_t))\,dt$$

Similarly, $\widehat{x}_{n+\frac{1}{2}}$ is the solution to the following ODE run for time $\alpha h$, starting at $\widehat{x}_n$ at time $t_n$:

$$d\widehat{x}_t = (\widehat{x}_t + \widehat{s}_{t_n}(\widehat{x}_n)) \, dt$$

So, we have

$$
\begin{aligned}
\partial_t \|x_t - \widehat{x}_t\|^2 &= 2\langle x_t - \widehat{x}_t, \partial_t x_t - \partial_t \widehat{x}_t \rangle \\
&= 2 \left( \|x_t - \widehat{x}_t\|^2 + \langle x_t - \widehat{x}_t, \nabla \ln q_t(x_t) - \widehat{s}_{t_n}(\widehat{x}_n) \rangle \right) \\
&\leq \left(2 + \frac{1}{h}\right) \|x_t - \widehat{x}_t\|^2 + h \|\nabla \ln q_t(x_t) - \widehat{s}_{t_n}(\widehat{x}_n)\|^2
\end{aligned}
$$

where the last line is by Young's inequality.

So, by Grönwall's inequality,

$$
\begin{aligned}
\|x_n^*(\alpha h) - \widehat{x}_{n+\frac{1}{2}}\|^2 &\leq \exp\left(\left(2 + \frac{1}{h}\right) \cdot \alpha h\right) \int_0^h h \|\nabla \ln q_{t_n - s}(x_n^*(s)) - \widehat{s}_{t_n}(\widehat{x}_n)\|^2 \, ds \\
&\lesssim h \int_0^h \|\nabla \ln q_{t_n - s}(x_n^*(s)) - \widehat{s}_{t_n}(\widehat{x}_n)\|^2 \, ds \\
&\lesssim h \int_0^h \|\nabla \ln q_{t_n - s}(x_n^*(s)) - \nabla \ln q_{t_n}(x_{t_n})\|^2 + \|\nabla \ln q_{t_n}(x_{t_n}) - \widehat{s}_{t_n}(x_{t_n})\|^2 \\
&\qquad + \|\widehat{s}_{t_n}(\widehat{x}_n) - \widehat{s}_{t_n}(x_{t_n})\|^2 \, ds.
\end{aligned}
$$

Taking expectation of the above quantity (as always over the intial choice of $\widehat{x}_0 \sim q_{t_0}$), we can now use a similar computation as the one in Equation (17):

$$
\begin{aligned}
\mathbb{E} \|x_n^*(\alpha h) - \widehat{x}_{n+\frac{1}{2}}\|^2 &\lesssim h \int_0^h \mathbb{E} \|\nabla \ln q_{t_n - s}(x_n^*(s)) - \nabla \ln q_{t_n}(x_{t_n})\|^2 \, ds \\
&\qquad + h^2 \mathbb{E} \|\nabla \ln q_{t_n}(x_{t_n}) - \widehat{s}_{t_n}(x_{t_n})\|^2 + h^2 \mathbb{E} \|\widehat{s}_{t_n}(\widehat{x}_n) - \widehat{s}_{t_n}(x_{t_n})\|^2 \\
&\lesssim h \int_0^h \mathbb{E} \|\nabla \ln q_{t_n - s}(x_n^*(s)) - \nabla \ln q_{t_n}(x_{t_n})\|^2 \, ds + h^2 \epsilon_{sc}^2 + h^2 L^2 \|\widehat{x}_n - x_{t_n}\|^2.
\end{aligned}
$$

Now we just need to bound $\mathbb{E} \|\nabla \ln q_{t_n - s}(x_n^*(s)) - \nabla \ln q_{t_n}(x_{t_n})\|^2$. By Corollary E.1,

$$\mathbb{E} \|\nabla \ln q_{t_n - s}(x_{t_n - s}) - \nabla \ln q_{t_n}(x_{t_n})\|^2 \lesssim L^2 d h^2 \left(L \vee \frac{1}{t_n}\right).$$

By Lipschitzness of $\nabla \ln q_t$ and Lemma B.1, for $s \leq h_n \leq 1/L$,

$$
\begin{aligned}
\|\nabla \ln q_{t_n - s}(x_n^*(s)) - \nabla \ln q_{t_n - s}(x_{t_n - s})\|^2 &\leq L^2 \|x_n^*(s) - x_{t_n - s}\|^2 \\
&\lesssim L^2 \exp\left(O(Lh_n)\right) \|\widehat{x}_n - x_{t_n}\|^2 \\
&\lesssim L^2 \|\widehat{x}_n - x_{t_n}\|^2.
\end{aligned}
$$

Combining the above two equations together, we get

$$\mathbb{E} \|\nabla \ln q_{t_n - s}(x_n^*(s)) - \nabla \ln q_{t_n}(x_{t_n})\|^2 \lesssim L^2 d h^2 \left(L \vee \frac{1}{t_n}\right) + L^2 \|\widehat{x}_n - x_{t_n}\|^2,$$

and thus

$$\mathbb{E} \|x_n^*(\alpha h) - \widehat{x}_{n+\frac{1}{2}}\|^2 \lesssim h^2 \epsilon_{\mathrm{sc}}^2 + L^2 d h^4 \left(L \vee \frac{1}{t_n}\right) + L^2 h^2 \mathbb{E} \|\widehat{x}_n - x_{t_n}\|^2.$$

Plugging the above bound into Equation (17) yields the claim. $\qquad \square$

**Lemma B.3** (Sequential Predictor Bias). *Suppose $L \geq 1$. In Algorithm 4, for all $n \in \{0, \ldots, N - 1\}$, let $x_n^*(t)$ be the solution of the true ODE starting at $\widehat{x}_n$ at time $t_n$ and running until time $t_n - t$, and let $x_t \sim q_t$ be the solution of the true ODE, starting at $\widehat{x}_0 \sim q_{t_0}$. If $h_n \lesssim \frac{1}{L}$, and $t_n - h_n \geq t_n/2$, we have*

$$\mathbb{E} \left\| \underset{\alpha}{\mathbb{E}} \widehat{x}_{n+1} - x_n^*(h_n) \right\|^2 \lesssim h_n^2 \epsilon_{\mathrm{sc}}^2 + L^4 d h_n^6 \left(L \vee \frac{1}{t_n}\right) + L^2 h_n^2 \mathbb{E} \|\widehat{x}_n - x_{t_n}\|^2$$

*where $\mathbb{E}_\alpha$ is the expectation with respect to the $\alpha$ chosen in the $n^{th}$ step, and $\mathbb{E}$ is the expectation with respect to the choice of the initial $\widehat{x}_0 \sim q_{t_0}$.*

*Proof.* For the proof, we wil fix $n$, and let $h := h_n$. By the integral formulation of the true ODE,

$$x_n^*(h) = e^h \widehat{x}_n + \int_{t_n-h}^{t_n} e^{s-(t_n-h)} \nabla \ln q_s(x_n^*(t_n - s)) \, \mathrm{d}s.$$

Thus, we have

$$\| \underset{\alpha}{\mathbb{E}} \, \widehat{x}_{n+1} - x_n^*(h) \|^2 = \| h \underset{\alpha}{\mathbb{E}} \, e^{(1-\alpha)h} \widehat{s}_{t_n - \alpha h}(\widehat{x}_{n+\frac{1}{2}}) - \int_{t_n-h}^{t_n} e^{s-(t_n-h)} \nabla \ln q_s(x_n^*(t_n - s)) \, \mathrm{d}s \|^2$$

$$\lesssim \underset{\alpha}{\mathbb{E}} \, \| h e^{(1-\alpha)h} \widehat{s}_{t_n - \alpha h}(\widehat{x}_{n+\frac{1}{2}}) - h e^{(1-\alpha)h} \nabla \ln q_{t_n - \alpha h}(x_n^*(\alpha h)) \|^2$$

$$+ \| h \cdot \underset{\alpha}{\mathbb{E}} \, e^{(1-\alpha)h} \nabla \ln q_{t_n - \alpha h}(x_n^*(\alpha h)) - \int_{t_n-h}^{t_n} e^{s-(t_n-h)} \nabla \ln q_s(x_n^*(t_n - s)) \, \mathrm{d}s \|^2 \,.$$

The second term is $0$ since

$$h \underset{\alpha}{\mathbb{E}} \, e^{(1-\alpha)h} \nabla \ln q_{t_n - \alpha h}(x_n^*(\alpha h)) = h \int_0^1 e^{(1-\alpha)h} \nabla \ln q_{t_n - \alpha h}(x_n^*(\alpha h)) \, \mathrm{d}\alpha$$

$$= \int_{t_n-h}^{t_n} e^{s-(t_n-h)} \nabla \ln q_s(x_n^*(t_n - s)) \, \mathrm{d}s \,.$$

For the first term, we have, by Lemma B.2

$$\mathbb{E} \, \| h e^{(1-\alpha)h} \widehat{s}_{t_n - \alpha h}(\widehat{x}_{n+\frac{1}{2}}) - h e^{(1-\alpha)h} \nabla \ln q_{t_n - \alpha h}(x_n^*(\alpha h)) \|^2$$

$$\lesssim h^2 \epsilon_{\mathrm{sc}}^2 + L^4 d h^6 \left( L \vee \frac{1}{t_n} \right) + L^2 h^2 \, \mathbb{E} \, \| \widehat{x}_n - x_{t_n} \|^2 \,.$$

The claimed bound follows. $\qquad\square$

**Lemma B.4** (Sequential Predictor Variance). *Suppose $L \geq 1$. In Algorithm 4, for all $n \in \{0, \dots, N-1\}$, let $x_n^*(t)$ be the solution of the true ODE starting at $\widehat{x}_n$ at time $t_n$ and running until time $t_n - t$, and let $x_t \sim q_t$ be the solution of the true ODE starting at $\widehat{x}_0 \sim q_{t_0}$. If $h_n \lesssim \frac{1}{L}$ and $t_n - h_n \geq t_n/2$, we have*

$$\mathbb{E} \, \| \widehat{x}_{n+1} - x_n^*(h_n) \|^2 \lesssim h_n^2 \epsilon_{\mathrm{sc}}^2 + L^2 d h_n^4 \left( L \vee \frac{1}{t_n} \right) + L^2 h_n^2 \, \mathbb{E} \, \| x_{t_n} - \widehat{x}_n \|^2$$

*where $\mathbb{E}$ refers to the expectation wrt the random $\alpha$ in the $n^{th}$ step, along with the initial choice $\widehat{x}_0 \sim q_{t_0}$.*

*Proof.* Fix $n$ and let $h := h_n$. We have

$$\mathbb{E} \, \| \widehat{x}_{n+1} - x_n^*(h) \|^2$$

$$= \mathbb{E} \, \| h \cdot e^{(1-\alpha)h} \widehat{s}_{t_n - \alpha h}(\widehat{x}_{n+\frac{1}{2}}) - \int_{t_n-h}^{t_n} e^{s-(t_n-h)} \nabla \ln q_s(x_n^*(t_n - s)) \, \mathrm{d}s \|^2$$

$$\lesssim \mathbb{E} \, \| h \cdot e^{(1-\alpha)h} \widehat{s}_{t_n - \alpha h}(\widehat{x}_{n+\frac{1}{2}}) - h e^{(1-\alpha)h} \nabla \ln q_{t_n - \alpha h}(x_n^*(\alpha h)) \|^2$$

$$+ \mathbb{E} \, \| h \cdot e^{(1-\alpha)h} \nabla \ln q_{t_n - \alpha h}(x_n^*(\alpha h)) - \int_{t_n-h}^{t_n} e^{(1-\alpha)h} \nabla \ln q_s(x_n^*(t_n - s)) \, \mathrm{d}s \|^2$$

$$+ \mathbb{E} \, \| \int_{t_n-h}^{t_n} e^{(1-\alpha)h} \nabla \ln q_s(x_n^*(t_n - s)) \, \mathrm{d}s - \int_{t_n-h}^{t_n} e^{s-(t_n-h)} \nabla \ln q_s(x_n^*(t_n - s)) \, \mathrm{d}s \|^2$$

The first term was bounded in Lemma B.2:

$$\mathbb{E} \, \| h \cdot e^{(1-\alpha)h} \widehat{s}_{t_n - \alpha h}(\widehat{x}_{n+\frac{1}{2}}) - h e^{(1-\alpha)h} \nabla \ln q_{t_n - \alpha h}(x_{t_n - \alpha h}) \|^2$$

$$\lesssim h^2 \epsilon_{\mathrm{sc}}^2 + L^4 d h^6 \left( L \vee \frac{1}{t_n} \right) + L^2 h^2 \, \mathbb{E} \, \| \widehat{x}_n - x_{t_n} \|^2 \,.$$

For the second term,

$$\mathbb{E} \left\| h \cdot e^{(1-\alpha)h} \nabla \ln q_{t_n - \alpha h}(x_n^*(\alpha h)) - \int_{t_n - h}^{t_n} e^{(1-\alpha)h} \nabla \ln q_s(x_n^*(t_n - s)) \, \mathrm{d}s \right\|^2$$

$$= \mathbb{E} \left\| \int_{t_n - h}^{t_n} e^{(1-\alpha)h} \cdot \left( \nabla \ln q_{t_n - \alpha h}(x_n^*(\alpha h)) - \nabla \ln q_s(x_n^*(t_n - s)) \right) \, \mathrm{d}s \right\|^2$$

$$\lesssim h \int_{t_n - h}^{t_n} \mathbb{E} \left\| \nabla \ln q_{t_n - \alpha h}(x_n^*(\alpha h)) - \nabla \ln q_s(x_n^*(t_n - s)) \right\|^2 \, \mathrm{d}s.$$

Now,

$$\mathbb{E} \left\| \nabla \ln q_{t_n - \alpha h}(x_n^*(\alpha h)) - \nabla \ln q_s(x_n^*(t_n - s)) \right\|^2 \lesssim \mathbb{E} \left\| \nabla \ln q_{t_n - \alpha h}(x_{t_n - \alpha h}) - \nabla \ln q_s(x_s) \right\|^2$$

$$+ \mathbb{E} \left\| \nabla \ln q_{t_n - \alpha h}(x_{t_n - \alpha h}) - \nabla \ln q_{t_n - \alpha h}(x_n^*(\alpha h)) \right\|^2$$

$$+ \mathbb{E} \left\| \nabla \ln q_s(x_s) - \nabla \ln q_s(x_n^*(t_n - s)) \right\|^2$$

The first of these terms is bounded in Corollary E.1:

$$\mathbb{E} \left\| \nabla \ln q_{t_n - \alpha h}(x_{t_n - \alpha h}) - \nabla \ln q_s(x_s) \right\|^2 \lesssim L^2 d h^2 \left( L \vee \frac{1}{t_n} \right)$$

For the remaining two terms, note that by the Lipschitzness of $\nabla \ln q_t$ and Lemma B.1,

$$\mathbb{E} \left\| \nabla \ln q_{t_n - \alpha h}(x_{t_n - \alpha h}) - \nabla \ln q_{t_n - \alpha h}(x_n^*(\alpha h)) \right\|^2 \leq L^2 \mathbb{E} \left\| x_{t_n - \alpha h} - x_n^*(\alpha h) \right\|^2$$

$$\lesssim L^2 \exp(O(Lh)) \mathbb{E} \left\| x_{t_n} - \widehat{x}_n \right\|^2$$

$$\lesssim L^2 \mathbb{E} \left\| x_{t_n} - \widehat{x}_n \right\|^2$$

and similarly, for $t_n - h \leq s \leq t_n$,

$$\mathbb{E} \left\| \nabla \ln q_s(x_s) - \nabla \ln q_s(x_n^*(t_n - s)) \right\|^2 \lesssim L^2 \mathbb{E} \left\| x_{t_n} - \widehat{x}_n \right\|^2. \tag{19}$$

Thus, we have shown that the second term in our bound on $\mathbb{E} \left\| \widehat{x}_{n+1} - x_n^*(h) \right\|^2$ is bounded as follows:

$$\mathbb{E} \left\| h \cdot e^{(1-\alpha)h} \nabla \ln q_{t_n - \alpha h}(x_n^*(\alpha h)) - \int_{t_n - h}^{t_n} e^{(1-\alpha)h} \nabla \ln q_s(x_n^*(t_n - s)) , \mathrm{d}s \right\|^2$$

$$\lesssim L^2 d h^4 \left( L \vee \frac{1}{t_n} \right) + L^2 h^2 \mathbb{E} \left\| x_{t_n} - \widehat{x}_n \right\|^2.$$

For the third term,

$$\mathbb{E} \left\| \int_{t_n - h}^{t_n} e^{(1-\alpha)h} \nabla \ln q_s(x_n^*(t_n - s)) , \mathrm{d}s - \int_{t_n - h}^{t_n} e^{s - (t_n - h)} \nabla \ln q_s(x_n^*(t_n - s)) , \mathrm{d}s \right\|^2$$

$$= \mathbb{E} \left\| \int_{t_n - h}^{t_n} \left( e^{(1-\alpha)h} - e^{s - (t_n - h)} \right) \nabla \ln q_s(x_n^*(t_n - s)) , \mathrm{d}s \right\|^2$$

$$\lesssim h \int_{t_n - h}^{t_n} \mathbb{E}_\alpha \left( e^{(1-\alpha)h} - e^{s - (t_n - h)} \right)^2 \mathbb{E}_{\widehat{x}_0 \sim q_{t_0}} \left\| \nabla \ln q_s(x_n^*(t_n - s)) \right\|^2 \, \mathrm{d}s.$$

Now, we have

$$\mathbb{E} \left\| \nabla \ln q_s(x_n^*(t_n - s)) \right\|^2 \lesssim \mathbb{E} \left\| \nabla \ln q_s(x_s) \right\|^2 + \mathbb{E} \left\| \nabla \ln q_s(x_n^*(t_n - s)) - \nabla \ln q_s(x_s) \right\|^2$$

$$\lesssim \frac{d}{s} + L^2 \mathbb{E} \left\| x_{t_n} - \widehat{x}_n \right\|^2.$$

where the last step follows by Lemma E.4 and equation 19. So,

$$\mathbb{E} \left\| \int_{t_n - h}^{t_n} e^{(1-\alpha)h} \nabla \ln q_s(x_n^*(t_n - s)) \, \mathrm{d}s - \int_{t_n - h}^{t_n} e^{s - (t_n - h)} \nabla \ln q_s(x_n^*(t_n - s)) \, \mathrm{d}s \right\|^2$$

$$\lesssim h \int_{t_n - h}^{t_n} \mathbb{E}_\alpha \left( e^{(1-\alpha)h} - e^{s - (t_n - h)} \right)^2 \cdot \left( \frac{d}{s} + L^2 \mathbb{E} \left\| x_{t_n} - \widehat{x}_n \right\|^2 \right) \, \mathrm{d}s$$

$$\lesssim h^4 \cdot \left( \frac{d}{t_n} + L^2 \mathbb{E} \left\| x_{t_n} - \widehat{x}_n \right\|^2 \right)$$

Thus, noting that $h \leq \frac{1}{L}$, we obtain the claimed bound on $\mathbb{E} \left\| \widehat{x}_{n+1} - x_n^*(h_n) \right\|^2$. $\qquad \square$

Finally, we put together the bias and variance bounds above to obtain a bound on the Wasserstein error at the end of the Predictor Step.

**Lemma B.5** (Sequential Predictor Wasserstein Guarantee). *Suppose that $L \geq 1$, and that for our sequence of step sizes $h_0, \dots, h_{N-1}$, $\sum_i h_i \leq 1/L$. Let $h_{max} = \max_i h_i$. Then, at the end of Algorithm 4,*

1. *If $t_N \gtrsim 1/L$,*

$$W_2^2(\widehat{x}_N, x_{t_N}) \lesssim \frac{\epsilon_{sc}^2}{L^2} + L^3 dh_{\max}^4 + L^2 dh_{\max}^3$$

2. *If $t_N \lesssim 1/L$ and $h_n \lesssim \frac{t_n}{2}$ for each $n$,*

$$W_2^2(\widehat{x}_N, x_{t_N}) \lesssim \frac{\epsilon_{sc}^2}{L^2} + \left(L^3 dh_{\max}^4 + L^2 dh_{\max}^3\right) \cdot N$$

*Here, $x_{t_N} \sim q_{t_N}$ is the solution of the true ODE beginning at $x_{t_0} = \widehat{x}_0 \sim q_{t_0}$.*

*Proof.* For all $n \in [1, \dots, N]$, let $y_n$ be the solution of the exact one step ODE starting from $\widehat{x}_{n-1}$. Let the operator $\mathbb{E}_\alpha$ be the expectation over the random choice of $\alpha$ in the $n^{th}$ iteration. Note that only $\widehat{x}_N$ depends on $\alpha$. We have

$$\mathbb{E}_\alpha \left[ \|x_{t_N} - \widehat{x}_N\|^2 \right] = \mathbb{E}_\alpha \left[ \|(x_{t_N} - y_N) - (\widehat{x}_N - y_N)\|^2 \right]$$

$$= \|x_{t_N} - y_N\|^2 - 2\langle x_{t_N} - y_N, \mathbb{E}_\alpha \widehat{x}_N - y_N\rangle + \mathbb{E}_\alpha \|\widehat{x}_N - y_N\|^2$$

$$\leq \left(1 + \frac{Lh_{N-1}}{2}\right) \|x_{t_N} - y_N\|^2 + \frac{2}{Lh_{N-1}} \| \mathbb{E}_\alpha \widehat{x}_N - y_N\|^2 + \mathbb{E}_\alpha \|\widehat{x}_N - y_N\|^2$$

$$\leq \exp\left(O(Lh_{N-1})\right) \|x_{t_{N-1}} - \widehat{x}_{N-1}\|^2 + \frac{2}{Lh_{N-1}} \| \mathbb{E}_\alpha \widehat{x}_N - y_N\|^2 + \mathbb{E}_\alpha \|\widehat{x}_N - y_N\|^2,$$

where the third line is by Young's inequality, and the fourth line is by Lemma B.1. Taking the expectation wrt $\widehat{x}_0 \sim q_{t_0}$, by Lemmas B.3 and B.4,

$$\mathbb{E} \|x_{t_N} - \widehat{x}_N\|^2$$

$$\leq \exp\left(O(Lh_{N-1})\right) \mathbb{E} \|x_{t_{N-1}} - \widehat{x}_{N-1}\|^2 + \frac{2}{Lh_{N-1}} \mathbb{E} \| \mathbb{E}_\alpha \widehat{x}_N - y_N\|^2 + \mathbb{E} \|\widehat{x}_N - y_N\|^2$$

$$\leq \exp\left(O(Lh_{N-1})\right) \mathbb{E} \|x_{t_{N-1}} - \widehat{x}_{N-1}\|^2$$

$$+ O\left(\frac{1}{Lh_{N-1}} \left(h_{N-1}^2 \epsilon_{sc}^2 + L^4 dh_{N-1}^6 \left(L \vee \frac{1}{t_{N-1}}\right) + L^2 h_{N-1}^2 \mathbb{E} \|x_{t_{N-1}} - \widehat{x}_{N-1}\|^2\right)\right)$$

$$+ O\left(h_{N-1}^2 \epsilon_{sc}^2 + L^2 dh_{N-1}^4 \left(L \vee \frac{1}{t_{N-1}}\right) + L^2 h_{N-1}^2 \mathbb{E} \|x_{t_{N-1}} - \widehat{x}_{N-1}\|^2\right)$$

$$\leq \exp\left(O(Lh_{N-1})\right) \mathbb{E} \|x_{t_{N-1}} - \widehat{x}_{N-1}\|^2$$

$$+ O\left(\frac{h_{N-1}\epsilon_{sc}^2}{L} + h_{N-1}^2 \epsilon_{sc}^2 + \left(L^3 dh_{N-1}^5 + L^2 dh_{N-1}^4\right) \left(L \vee \frac{1}{t_{N-1}}\right)\right)$$

By induction, noting that $x_{t_0} = \widehat{x}_0$, we have

$$\mathbb{E} \|x_{t_N} - \widehat{x}_N\|^2 \lesssim \sum_{n=0}^{N-1} \left(\frac{h_n \epsilon_{sc}^2}{L} + h_n^2 \epsilon_{sc}^2 + \left(L^3 dh_n^5 + L^2 dh_n^4\right) \cdot \left(L \vee \frac{1}{t_n}\right)\right) \cdot \exp\left(O\left(L \sum_{i=n+1}^{N-1} h_i\right)\right).$$

By assumption, $\sum_i h_i \leq \frac{1}{L}$. In the first case, $L \vee \frac{1}{t_n} \lesssim L$ for all $n$, so

$$W_2^2(\widehat{x}_N, x_{t_N}) \lesssim \frac{\epsilon_{sc}^2}{L^2} + L^3 dh_{\max}^4 + L^2 dh_{\max}^3.$$

In the second case,

$$W_2^2(\widehat{x}_N, x_{t_N}) \lesssim \frac{\epsilon_{sc}^2}{L^2} + \left(L^3 dh_{\max}^4 + L^2 dh_{\max}^3\right) \cdot \sum_{n=0}^{N-1} \frac{h_n}{t_n}$$

$$\lesssim \frac{\epsilon_{sc}^2}{L^2} + \left(L^3 dh_{\max}^4 + L^2 dh_{\max}^3\right) \cdot N.$$ $\qquad\square$

## B.2 CORRECTOR STEP

For the sequential algorithm, we make use of the underdamped Langevin corrector step and analysis from (Chen et al., 2023b). We reproduce the same here for convenience.

The underdamped Langevin Monte Carlo process with step size $h$ is given by:

$$\begin{aligned} \mathrm{d}\widehat{x}_t &= \widehat{v}_t\,\mathrm{d}t \\ \mathrm{d}\widehat{v}_t &= (\widehat{s}(\widehat{x}_{\lfloor t/h\rfloor h}) - \gamma\widehat{v}_t)\,\mathrm{d}t + \sqrt{2\gamma}\,\mathrm{d}B_t \end{aligned} \tag{20}$$

where $B_t$ is Brownian motion, and $\widehat{s}$ satisfies

$$\mathbb{E}_{x\sim q}\left\|\widehat{s}(x) - \nabla\ln q(x)\right\|^2 \le \epsilon_{\mathrm{sc}}^2\,. \tag{21}$$

for some target measure $q$. Here, we set the friction parameter $\gamma = \Theta(\sqrt{L})$.

Then, our corrector step is as follows.

---

**Algorithm 5** CORRECTORSTEP (SEQUENTIAL)

---

**Input parameters:**

- Starting sample $\widehat{x}_0$, Total time $T_{\mathrm{corr}}$, Step size $h_{\mathrm{corr}}$, Score estimate $\widehat{s}$

1. Run underdamped Langevin Monte Carlo in equation 20 for total time $T_{\mathrm{corr}}$ using step size $h_{\mathrm{corr}}$, and let the result be $\widehat{x}_N$.
2. Return $\widehat{x}_N$.

---

**Theorem B.6** (Theorem 5 of (Chen et al., 2023b), restated). *Suppose Eq. equation 21 holds. For any distribution $p$ over $\mathbb{R}^d$, and total time $T_{\mathrm{corr}} \lesssim 1/\sqrt{L}$, if we let $p_N$ be the distribution of $\widehat{x}_N$ resulting from running Algorithm 5 initialized at $\widehat{x}_0 \sim p$, then we have*

$$\mathsf{TV}(p_N, q) \lesssim \frac{W_2(p,q)}{L^{1/4}T_{\mathrm{corr}}^{3/2}} + \frac{\epsilon_{\mathrm{sc}}T_{\mathrm{corr}}^{1/2}}{L^{1/4}} + L^{3/4}T_{\mathrm{corr}}^{1/2}d^{1/2}h_{\mathrm{corr}}\,.$$

**Corollary B.7** (Underdamped Corrector). *For $T_{\mathrm{corr}} = \Theta\left(\frac{1}{\sqrt{L}d^{1/18}}\right)$*

$$\mathsf{TV}(p_N, q) \lesssim W_2(p,q) \cdot d^{1/12} \cdot \sqrt{L} + \frac{\epsilon_{\mathrm{sc}}}{\sqrt{L}d^{1/36}} + \sqrt{L}d^{17/36}h_{\mathrm{corr}}\,.$$

## B.3 END-TO-END ANALYSIS

Finally, we put together the analysis of the predictor and corrector step to obtain our final $\widetilde{O}(d^{5/12})$ dependence on sampling time. We first show that carefully choosing the amount of time to run the corrector results in small TV error after successive rounds of the predictor and corrector steps in Lemma B.8. Finally, we iterate this bound to obtain our final guarantee, given by Theorem B.10.

**Lemma B.8** (TV error after one round of predictor and corrector). *Let $x_t \sim q_t$ be a sample from the true distribution at time $t$. Let $t_n = T - n/L$ for $n \in [0,\dots,N_0]$. If we set $T_{\mathrm{corr}} = \Theta\left(\frac{1}{\sqrt{L}d^{1/18}}\right)$, we have,*

*1. For $n \in [0,\dots,N_0-1]$, if $t_n \gtrsim 1/L$,*

$$\mathsf{TV}(\widehat{x}_{n+1}, x_{t_{n+1}}) \le \mathsf{TV}(\widehat{x}_n, x_{t_n}) + O\left(L^2 d^{7/12}h_{\mathrm{pred}}^2 + L^{3/2}d^{7/12}h_{\mathrm{pred}}^{3/2} + \sqrt{L}d^{17/36}h_{\mathrm{corr}} + \frac{\epsilon_{\mathrm{sc}}d^{1/12}}{\sqrt{L}}\right)$$

*2. If $t_{N_0} \lesssim 1/L$,*

$$\mathsf{TV}(\widehat{x}_{N_0+1}, x_\delta) \le \mathsf{TV}(\widehat{x}_{N_0}, x_{t_{N_0}})$$
$$+ O\left(\left(L^2 d^{7/12}h_{\mathrm{pred}}^2 + L^{3/2}d^{7/12}h_{\mathrm{pred}}^{3/2}\right) \cdot \sqrt{\log\frac{h_{\mathrm{pred}}}{\delta}} + \sqrt{L}d^{17/36}h_{\mathrm{corr}} + \frac{\epsilon_{\mathrm{sc}}d^{1/12}}{\sqrt{L}}\right)$$

---

**Algorithm 6** SEQUENTIALALGORITHM

---

**Input parameters:**

- Start time $T$, End time $\delta$, Corrector steps time $T_{\text{corr}} \lesssim 1/\sqrt{L}$, Number of predictor-corrector steps $N_0$, Predictor step size $h_{\text{pred}}$, Corrector step size $h_{\text{corr}}$, Score estimates $\widehat{s}_t$

1. Draw $\widehat{x}_0 \sim \mathcal{N}(0, I_d)$.

2. For $n = 0, \ldots, N_0 - 1$:

   (a) Starting from $\widehat{x}_n$, run Algorithm 4 with starting time $T - n/L$ using step sizes $h_{\text{pred}}$ for all $N$ steps, with $N = \frac{1}{Lh_{\text{pred}}}$, so that the total time is $1/L$. Let the result be $\widehat{x}'_{n+1}$.

   (b) Starting from $\widehat{x}'_{n+1}$, run Algorithm 5 for total time $T_{\text{corr}}$ with step size $h_{\text{corr}}$ and score estimate $\widehat{s}_{T-(n+1)/L}$ to obtain $\widehat{x}_{n+1}$.

3. Starting from $\widehat{x}_{N_0}$, run Algorithm 4 with starting time $T - N_0/L$ using step sizes $h_{\text{pred}}/2, h_{\text{pred}}/4, h_{\text{pred}}/8, \ldots, \delta$ to obtain $\widehat{x}'_{N_0+1}$.

4. Starting from $\widehat{x}'_{N_0+1}$, run Algorithm 5 for total time $T_{\text{corr}}$ with step size $h_{\text{corr}}$ and score estimate $\widehat{s}_\delta$ to obtain $\widehat{x}_{N_0+1}$.

5. Return $\widehat{x}_{N_0+1}$.

---

*Proof.* For $n \in [0, \ldots, N_0]$, let $\widehat{y}_{n+1}$ be the result of a single predictor-corrector sequence as described in step 2 of Algorithm 6, but starting from $x_{t_n} \sim q_{t_n}$ instead of $\widehat{x}_n$. Additionally, let $\widehat{y}_{N_0+1}$ be the result of running steps 3 and 4 starting from $x_{t_{N_0}} \sim q_{t_{N_0}}$ instead of $\widehat{x}_{N_0}$. Similarly, let $\widehat{y}'_{n+1}$ be the result of only applying the predictor step starting from $x_{t_n} \sim q_{t_n}$, analogous to $\widehat{x}'_{n+1}$ defined in step 2a.

We have, by the triangle inequality and the data-processing inequality, for $n \in [0, \ldots, N_0 - 1]$,

$$\mathsf{TV}(\widehat{x}_{n+1}, x_{t_{n+1}}) \leq \mathsf{TV}(\widehat{x}_{n+1}, \widehat{y}_{n+1}) + \mathsf{TV}(\widehat{y}_{n+1}, x_{t_{n+1}})$$
$$\leq \mathsf{TV}(\widehat{x}_n, x_{t_n}) + \mathsf{TV}(\widehat{y}_{n+1}, x_{t_{n+1}})$$

By Corollary B.7,

$$\mathsf{TV}(\widehat{y}_{n+1}, x_{t_{n+1}}) \lesssim W_2(\widehat{y}'_{n+1}, x_{t_{n+1}}) \cdot d^{1/12} \cdot \sqrt{L} + \frac{\epsilon_{\text{sc}}}{\sqrt{L}d^{1/36}} + \sqrt{L}d^{17/36}h_{\text{corr}}$$

Now, for $t_n \gtrsim 1/L$, by Lemma B.5,

$$W_2(\widehat{y}'_{n+1}, x_{t_{n+1}}) \lesssim \frac{\epsilon_{\text{sc}}}{L} + L^{3/2}\sqrt{d}h_{\text{pred}}^2 + L\sqrt{d}h_{\text{pred}}^{3/2}.$$

Combining the above gives the first claim.

For the second claim, similar to above, we have

$$\mathsf{TV}(\widehat{x}_{N_0+1}, x_\delta) \leq \mathsf{TV}(\widehat{x}_{N_0+1}, \widehat{y}_{N_0+1}) + \mathsf{TV}(\widehat{y}_{N_0+1}, x_\delta)$$
$$\leq \mathsf{TV}(\widehat{x}_{N_0}, x_{t_{N_0}}) + \mathsf{TV}(\widehat{y}_{N_0+1}, x_\delta)$$

By Corollary B.7,

$$\mathsf{TV}(\widehat{y}_{N_0+1}, x_\delta) \leq W_2(\widehat{y}'_{N_0+1}, x_\delta) \cdot d^{1/12} \cdot \sqrt{L} + \frac{\epsilon_{\text{sc}}}{\sqrt{L}d^{1/36}} + \sqrt{L}d^{17/36}h_{\text{corr}}$$

For $t_{N_0} \lesssim 1/L$, by Lemma B.5, noting that the number of predictor steps in this case is $O\left(\log \frac{h_{\text{pred}}}{\delta}\right)$,

$$W_2(\widehat{y}'_{N_0+1}, x_\delta) \lesssim \frac{\epsilon_{\text{sc}}}{L} + \left(L^{3/2}\sqrt{d}h_{\text{pred}}^2 + L\sqrt{d}h_{\text{pred}}^{3/2}\right) \cdot \sqrt{\log \frac{h_{\text{pred}}}{\delta}}$$

The second claim follows by combining the above. $\square$

We recall the following lemma on the convergence of the OU process from (Chen et al., 2023b)

**Lemma B.9** (Lemma 13 of (Chen et al., 2023b)). *Let $q_t$ denote the marginal law of the OU process started at $q_0 = q$. Then, for all $T \gtrsim 1$,*

$$\mathsf{TV}(q_T, \mathcal{N}(0, I_d)) \lesssim (\sqrt{d} + \mathfrak{m}_2) \exp(-T)$$

Finally, we prove our main theorem on the convergence of our sequential algorithm.

**Theorem B.10** (Convergence bound for sequential algorithm). *Suppose Assumptions 2.1-2.4 hold. If $\widehat{x}$ denotes the output of Algorithm 6, for $T = \Theta\left(\log\left(\frac{d \vee \mathfrak{m}_2^2}{\epsilon^2}\right)\right), T_{\mathrm{corr}} = \Theta\left(\frac{1}{\sqrt{L}d^{1/18}}\right)$ and $\delta = \Theta\left(\frac{\epsilon^2}{L^2(d \vee \mathfrak{m}_2^2)}\right)$, if we set $h_{\mathrm{pred}} = \widetilde{\Theta}\left(\min\left(\frac{\epsilon^{1/2}}{d^{1/3}L^{3/2}}, \frac{\epsilon^{2/3}}{d^{5/12}L^{5/3}}\right) \cdot \frac{1}{\log(\mathfrak{m}_2)}\right), h_{\mathrm{corr}} = \widetilde{\Theta}\left(\frac{\epsilon}{d^{17/36}L^{3/2}\log(m_2)}\right)$, and if the score estimation satisfies $\epsilon_{\mathrm{sc}} \leq \widetilde{O}\left(\frac{\epsilon}{\sqrt{L}d^{1/12}\log \mathfrak{m}_2}\right)$, we have that*

$$\mathsf{TV}(\widehat{x}, x_0) \lesssim \epsilon$$

*with iteration complexity $\widetilde{\Theta}\left(\frac{L^{5/3}d^{5/12}}{\epsilon} \cdot \log^2(\mathfrak{m}_2)\right)$*

*Proof.* We will let $t_n = T - n/L$. First, note that by Lemma B.9

$$\mathsf{TV}(\widehat{x}_0, x_{t_0}) \lesssim (\sqrt{d} + \mathfrak{m}_2) \exp(-T)$$

We divide our analysis into two steps. For the first $N_0 = O(LT)$ steps, we iterate the first part of Lemma B.8 to obtain

$$\mathsf{TV}(\widehat{x}_{N_0}, x_{t_{N_0}}) \leq \mathsf{TV}(\widehat{x}_0, x_{t_0}) + O\left(L^2 d^{7/12}h_{\mathrm{pred}}^2 + L^{3/2}d^{7/12}h_{\mathrm{pred}}^{3/2} + \sqrt{L}d^{17/36}h_{\mathrm{corr}} + \frac{\epsilon_{\mathrm{sc}}d^{1/12}}{\sqrt{L}}\right) \cdot N_0$$

$$\lesssim \left(\sqrt{d} + \mathfrak{m}_2\right)\exp(-T) + L^3 d^{7/12}h_{\mathrm{pred}}^2 T + L^{5/2}d^{7/12}h_{\mathrm{pred}}^{3/2}T + L^{3/2}d^{17/36}h_{\mathrm{corr}}T + \epsilon_{\mathrm{sc}}d^{1/12}T\sqrt{L}$$

Applying the second part of Lemma B.8 for the second stage of the algorithm, we have

$$\mathsf{TV}(\widehat{x}_{N_0+1}, x_\delta)$$

$$\lesssim \left(\sqrt{d} + \mathfrak{m}_2\right)\exp(-T) + \left(L^3 d^{7/12}h_{\mathrm{pred}}^2 + L^{5/2}d^{7/12}h_{\mathrm{pred}}^{3/2}\right)\left(T + \sqrt{\log\frac{h_{\mathrm{pred}}}{\delta}}\right)$$

$$+ L^{3/2}d^{17/36}h_{\mathrm{corr}}T + \epsilon_{\mathrm{sc}}d^{1/12}T\sqrt{L}$$

Setting $T = \Theta\left(\log\left(\frac{d \vee \mathfrak{m}_2^2}{\epsilon^2}\right)\right), h_{\mathrm{pred}} = \widetilde{\Theta}\left(\min\left(\frac{\epsilon^{1/2}}{d^{1/3}L^{3/2}}, \frac{\epsilon^{2/3}}{d^{5/12}L^{5/3}}\right)\frac{1}{\log(\mathfrak{m}_2)}\right)$, and $h_{\mathrm{corr}} = \widetilde{\Theta}\left(\frac{\epsilon}{d^{17/36}L^{3/2}} \cdot \frac{1}{\log(\mathfrak{m}_2)}\right)$, if the score estimation error satisfies $\epsilon_{\mathrm{sc}} \leq \widetilde{O}\left(\frac{\epsilon}{\sqrt{L}d^{1/12}\log(\mathfrak{m}_2)}\right)$, with iteration complexity $\widetilde{\Theta}\left(\frac{L^{5/3}d^{5/12}\log^2 \mathfrak{m}_2}{\epsilon}\right)$, we obtain $\mathsf{TV}(\widehat{x}_{N_0+1}, x_\delta) \leq \epsilon$. $\square$

## C  PARALLEL ALGORITHM

### C.1  PREDICTOR STEP

In this section, we will apply a parallel version of randomized midpoint for the predictor step, where only $\widetilde{\Theta}(\log^2(\frac{Ld}{\epsilon}))$ iteration complexity will be required to attain our desired error bound for one predictor step.

In each iteration $n$, we will first sample $R$ randomized midpoints that are in expectation evenly spaced with $\delta_n = \frac{h_n}{R_n}$ time intervals between consecutive midpoints. Next, in our step (c), we provide an initial estimate on the $x$ value of midpoints using our estimate of position $\widehat{x}_n$ at time $t_n$ provided by iteration $n - 1$. This step is analogous to step (c) in Algorithm 4 for the sequential predictor step. Then, in step (d) we refine our initial estimates by using a discrete version of Picard iteration, where for round $k$, we compute a new estimate of $x_{t_n - \alpha_i h_n}$ based on the estimates of

$x_{t_n - \alpha_j h_n}$ for $j \leq i$ in round $k-1$. Note that a trajectory $x(t)$ that starts from time $t_0$ and follows the true ODE is a fix point of operator $\tau$ that maps continuous function to continuous function, where

$$\tau(x)(t) = e^{t-t_0} x(t_0) + \int_{t_0}^t e^{s-t_0} \cdot \nabla q_s(x(s)) \mathrm{d}s.$$

By smoothness of the true ODE, the continuous Picard iteration converges exponentially to the true trajectory, and we will show that discretization error for Picard iteration is controlled. After the refinements have sufficiently reduced the estimation error for our randomized midpoints, we make a final calculation, estimating the value of $x_{t_n + h_n}$ based on the estimated value at all the randomized midpoints.

---

**Algorithm 7** PREDICTORSTEP (PARALLEL)

---

**Input parameters:**

- Starting sample $\widehat{x}_0$, Starting time $t_0$, Number of steps $N$, Step size $\{h_n\}_{n=0}^{N-1}$, Number of midpoint estimates $\{R_n\}_{n=0}^{N-1}$, Number of parallel iteration $\{K_n\}_{n=0}^{N-1}$, Score estimates $\widehat{s}_t$
- For all $n = 0, \cdots, N-1$: let $\delta_n = \frac{h_n}{R_n}$

1. For $n = 0, \ldots, N-1$:
    (a) Let $t_n = t_0 - \sum_{w=0}^{n-1} h_w$
    (b) Randomly sample $\alpha_i$ uniformly from $[(i-1)/R_n, i/R_n]$ for all $i \in \{1, \cdots, R_n\}$
    (c) For $i = 1, \cdots, R_n$ in parallel: Let $\widehat{x}_{n,i}^{(0)} = e^{\alpha_i h_n} \widehat{x}_n + (e^{\alpha_i h_n} - 1) \cdot \widehat{s}_{t_n}(\widehat{x}_n)$
    (d) For $k = 1, \cdots, K_n$:
        For $i = 1, \cdots, R_n$ in parallel:
        $$\widehat{x}_{n,i}^{(k)} := e^{\alpha_i h_n} \widehat{x}_n + \sum_{j=1}^i \left( e^{\alpha_i h_n - (j-1)\delta_n} - \max(e^{\alpha_i h_n - j\delta_n}, 1) \right) \cdot \widehat{s}_{t_n - \alpha_j h_n}(\widehat{x}_{n,j}^{(k-1)})$$
    (e) $\widehat{x}_{n+1} = e^{h_n} \widehat{x}_n + \delta_n \cdot \sum_{i=1}^R e^{h_n - \alpha_i h_n} \widehat{s}_{t_n - \alpha_i h_n}(\widehat{x}_{n,i}^{(K_n)})$
2. Let $t_N = t_0 - \sum_{n=0}^{N-1} h_n$
3. Return $\widehat{x}_N, t_N$.

---

In our analysis, we follow the same notation as in Appendix B.1. We first establish a $\mathrm{poly}(L, d)$ bound on the initial estimation error incurred in step (c) of each iteration in Algorithm 7.

**Claim C.1.** *Suppose $L \geq 1$. Assume $h_n \lesssim 1/L$. For any $n = 0, \cdots, N-1$, suppose we draw $\widehat{x}_n$ from an arbitrary distribution $p_n$, then run step (a) - (e) in Algorithm 7. Then for any $i = 1, \cdots, R_n$,*

$$\mathbb{E} \left\| \widehat{x}_{n,i}^{(0)} - x_n^*(\alpha_i h_n) \right\|^2 \lesssim h_n^2 \epsilon_{\mathrm{sc}}^2 + L^2 d h_n^4 (L \vee \frac{1}{t_n - h_n}) + L^2 h_n^2 \left\| \widehat{x}_n - x_{t_n} \right\|^2,$$

*where $x_n^*(t)$ is solution of the true ODE starting at $\widehat{x}_n$ at time $t_n$ and running until time $t_n - t$.*

*Proof.* Notice that in step (c) of Algorithm 7, the initial estimate of the randomized midpoint is done with the exact same formula as in step (c) of Algorithm 6, except we calculate this initial estimate for $R_n$ different randomized midpoints. Notice also that the bound for discretization error in Theorem B.2 is not dependent on specific value of $\alpha$, as long as the randomed value is at most 1. Hence we can use identical calculation to yield the claim. $\qquad \square$

Next, we show how to drive the initialization error from Lemma C.1 down (exponentially) using the Picard iterations in step (d) of Algorithm 7.

**Lemma C.2.** *Suppose $L \geq 1$. Assume $h_n \lesssim 1/L$. For all iterations $n \in \{0, \cdots, N-1\}$, suppose we draw $\widehat{x}_n$ from an arbitrary distribution $p_n$, then run step (a) - (e) in Algorithm 7. Then for all*

$k \in \{1, \cdots K_n\}$ and $i \in \{1, \cdots, R_n\}$,

$$
\mathbb{E} \left\| \widehat{x}_{n,i}^{(k)} - x_n^*(\alpha_i h_n) \right\|^2 \lesssim \left( 8 h_n^2 L^2 \right)^k \cdot \left( \frac{1}{R} \sum_{j=1}^{R} \left\| \widehat{x}_{n,j}^{(0)} - x_n^*(\alpha_j h_n) \right\|^2 \right)
$$

$$
+ h_n^2 \left( \epsilon_{\mathrm{sc}}^2 + \frac{L^2 d h_n^2}{R_n^2} (L \vee \frac{1}{t_n - h_n}) + L^2 \cdot \mathbb{E} \left\| \widehat{x}_n - x_{t_n} \right\|^2 \right), \quad (22)
$$

where $x_n^*(t)$ is solution of the true ODE starting at $\widehat{x}_n$ at time $t_n$ and running until time $t_n - t$.

*Proof.* Fixing iteration $n$, we will let $h := h_n$, $R := R_n$ and $\delta := \delta_n$. The formula of $\widehat{x}_{n,i}^{(k)}$ and $x_n^*(\alpha_i h)$ has the same coefficient for $\hat{x}_n$, thus we can bound the difference as follows:

$$
\mathbb{E} \left\| \widehat{x}_{n,i}^{(k)} - x_n^*(\alpha_i h) \right\|^2
$$

$$
\leq \mathbb{E} \left\| \sum_{j=1}^{i} \left( \int_{t_n - \min(j\delta, \alpha_i h)}^{t_n - (j-1)\delta} e^{s - (t_n - \alpha_i h)} \, \mathrm{d}s \cdot \widehat{s}_{t_n - \alpha_j h} (\widehat{x}_{n,j}^{(k-1)}) - \int_{t_n - \alpha_i h}^{t_n} e^{s - (t_n - \alpha_i h)} \nabla \ln q_s (x_n^*(t_n - s)) \, \mathrm{d}s \right) \right\|^2
$$

$$
\lesssim \mathbb{E} \left\| \sum_{j=1}^{i} \int_{t_n - \min(j\delta, \alpha_i h)}^{t_n - (j-1)\delta} e^{s - (t_n - \alpha_i h)} \, \mathrm{d}s \cdot \left( \widehat{s}_{t_n - \alpha_j h} (\widehat{x}_{n,j}^{(k-1)}) - \nabla \ln q_{t_n - \alpha_j h} (x_n^*(\alpha_j h)) \right) \right\|^2 \quad (23)
$$

$$
+ \mathbb{E} \left\| \sum_{j=1}^{i} \int_{t_n - \min(j\delta, \alpha_i h)}^{t_n - (j-1)\delta} e^{s - (t_n - \alpha_i h)} \left( \nabla q_{t_n - \alpha_j h} (x_n^*(\alpha_j h)) - \nabla \ln q_s (x_n^*(t_n - s)) \right) \, \mathrm{d}s \right\|^2 .
$$

$$(24)$$

The first to second line is by definition and the second to third line is by Young's inequality. Now, we will bound Equation (23) and Equation (24) separately. By Theorem 2.2 and Theorem 2.4,

$$
\mathbb{E} \left\| \widehat{s}_{t_n - \alpha_j h} (\widehat{x}_{n,j}^{(k-1)}) - \nabla \ln q_{t_n - \alpha_j h} (x_n^*(\alpha_j h)) \right\|^2
$$

$$
\lesssim \mathbb{E} \left\| \widehat{s}_{t_n - \alpha_j h} (\widehat{x}_{n,j}^{(k-1)}) - s_{t_n - \alpha_j h} (x_{t_n - \alpha_j h}) \right\|^2 + \mathbb{E} \left\| \nabla \ln q_{t_n - \alpha_j h} (x_n^*(\alpha_j h)) - \nabla \ln q_{t_n - \alpha_j h} (x_{t_n - \alpha_j h}) \right\|^2
$$

$$
+ \mathbb{E} \left\| s_{t_n - \alpha_j h} (x_{t_n - \alpha_j h}) - \nabla \ln q_{t_n - \alpha_j h} (x_{t_n - \alpha_j h}) \right\|^2
$$

$$
\lesssim \epsilon_{\mathrm{sc}}^2 + L^2 \cdot \mathbb{E} \left\| \widehat{x}_{n,j}^{(k-1)} - x_{t_n - \alpha_j h} \right\|^2 + L^2 \mathbb{E} \left\| \widehat{x}_n^*(\alpha_j h) - x_{t_n - \alpha_j h} \right\|^2
$$

$$
\lesssim \epsilon_{\mathrm{sc}}^2 + L^2 \cdot \mathbb{E} \left\| \widehat{x}_{n,j}^{(k-1)} - x_n^*(\alpha_j h) \right\|^2 + L^2 \cdot \mathbb{E} \left\| x_n^*(\alpha_j h) - x_{t_n - \alpha_j h} \right\|^2 .
$$

By Theorem B.1, for any $s \in [0, h]$,

$$
\mathbb{E} \left\| x_n^*(s) - x_{t_n - s} \right\|^2 \lesssim \exp \left( O(Lh) \right) \cdot \mathbb{E} \left\| \widehat{x}_n - x_{t_n} \right\|^2 \lesssim \mathbb{E} \left\| \widehat{x}_n - x_{t_n} \right\|^2, \quad (25)
$$

thus $L^2 \cdot \mathbb{E} \left\| x_n^*(\alpha_j h) - x_{t_n - \alpha_j h} \right\|^2 \lesssim L^2 \mathbb{E} \left\| \widehat{x}_n - x_{t_n} \right\|^2$. The term in Equation (23) can now be bounded as follows

$$
\mathbb{E} \left\| \sum_{j=1}^i \int_{t_n - \min(j\delta, \alpha_i h)}^{t_n - (j-1)\delta} e^{s - (t_n - \alpha_i h)} \, \mathrm{d}s \cdot \left( \widehat{s}_{t_n - \alpha_j h}(\widehat{x}_{n,j}^{(k-1)}) - \nabla \ln q_{t_n - \alpha_j h}(x_n^*(\alpha_j h)) \right) \right\|^2
$$

$$
\leq R \cdot \sum_{j=1}^i \mathbb{E} \left\| \int_{t_n - \min(j\delta, \alpha_i h)}^{t_n - (j-1)\delta} e^{s - (t_n - \alpha_i h)} \, \mathrm{d}s \cdot \left( \widehat{s}_{t_n - \alpha_j h}(\widehat{x}_{n,j}^{(k-1)}) - \nabla \ln q_{t_n - \alpha_j h}(x_n^*(\alpha_j h)) \right) \right\|^2
$$

$$
\leq R \cdot \delta^2 \cdot e^{2\alpha_i h} \sum_{j=1}^i \mathbb{E} \left\| \widehat{s}_{t_n - \alpha_j h}(\widehat{x}_{n,j}^{(k-1)}) - \nabla \ln q_{t_n - \alpha_j h}(x_n^*(\alpha_j h)) \right\|^2
$$

$$
\leq 2R \cdot \delta^2 \cdot e^{2\alpha_i h} \sum_{j=1}^i \left( \epsilon_{\mathrm{sc}}^2 + L^2 \cdot \mathbb{E} \left\| \widehat{x}_{n,j}^{(k-1)} - x_n^*(\alpha_j h) \right\|^2 + L^2 \mathbb{E} \left\| \widehat{x}_n - x_{t_n} \right\|^2 \right)
$$

$$
\leq 2R^2 \cdot \delta^2 \cdot e^{2\alpha_i h} \frac{1}{R} \sum_{j=1}^R \left( \epsilon_{\mathrm{sc}}^2 + L^2 \cdot \mathbb{E} \left\| \widehat{x}_{n,j}^{(k-1)} - x_n^*(\alpha_j h) \right\|^2 + L^2 \mathbb{E} \left\| \widehat{x}_n - x_{t_n} \right\|^2 \right)
$$

$$
\leq 4h^2 \epsilon_{\mathrm{sc}}^2 + 4h^2 L^2 \mathbb{E} \left\| \widehat{x}_n - x_{t_n} \right\|^2 + 4h^2 L^2 \cdot \frac{1}{R} \sum_{j=1}^R \cdot \mathbb{E} \left\| \widehat{x}_{n,j}^{(k-1)} - x_n^*(\alpha_j h) \right\|^2.
$$

The first to second line is by inequality $(\sum_{i=1}^n a_i)^2 \leq n \sum_{i=1}^n a_i^2$, the second to third line is by the fact that $\int_{t_n - \min(j\delta, \alpha_i h)}^{t_n - (j-1)\delta} e^{s - (t_n - \alpha_i h)} \, \mathrm{d}s \leq \delta \cdot e^{\alpha_i h}$, the fifth to sixth line is by $R\delta = h$ and that $e^{2\alpha_i h}$ is at most 2 when $h < 1/4$.

Similarly, the term in Equation (24) can be bounded as follows

$$
\mathbb{E} \left\| \sum_{j=1}^i \int_{t_n - \min(j\delta, \alpha_i h)}^{t_n - (j-1)\delta} e^{s - (t_n - \alpha_i h)} \left( \nabla \ln q_{t_n - \alpha_j h}(x_n^*(\alpha_j h)) - \nabla \ln q_s(x_n^*(t_n - s)) \right) \, \mathrm{d}s \right\|^2
$$

$$
\leq R \cdot \sum_{j=1}^i \mathbb{E} \left\| \int_{t_n - \min(j\delta, \alpha_i h)}^{t_n - (j-1)\delta} e^{s - (t_n - \alpha_i h)} \left( \nabla \ln q_{t_n - \alpha_j h}(x_n^*(\alpha_j h)) - \nabla \ln q_s(x_n^*(t_n - s)) \right) \, \mathrm{d}s \right\|^2
$$

$$
\leq R \cdot \delta \cdot e^{2\alpha_i h} \cdot \sum_{j=1}^i \int_{t_n - \min(j\delta, \alpha_i h)}^{t_n - (j-1)\delta} \mathbb{E} \left\| \nabla \ln q_{t_n - \alpha_j h}(x_n^*(\alpha_j h)) - \nabla \ln q_s(x_n^*(t_n - s)) \right\|^2 \, \mathrm{d}s
$$

Since $|x_n^*(\alpha_j h) - s| \leq \delta$,

$$
\mathbb{E} \left\| \nabla q_{t_n - \alpha_j h}(x_n^*(\alpha_j h)) - \nabla \ln q_s(x_n^*(t_n - s)) \right\|^2
$$

$$
\leq 3\mathbb{E} \left\| \nabla q_{t_n - \alpha_j h}(x_n^*(\alpha_j h)) - \nabla q_{t_n - \alpha_j h}(x_{t_n - \alpha_j h}) \right\|^2 + 3\mathbb{E} \left\| \nabla q_s(x_s) - \nabla q_s(x_n^*(t_n - s)) \right\|^2
$$

$$
+ 3\mathbb{E} \left\| \nabla q_{t_n - \alpha_j h}(x_{t_n - \alpha_j h}) - \nabla q_s(x_s) \right\|^2
$$

$$
= 3\mathbb{E} \left\| \nabla q_{t_n - \alpha_j h}(x_n^*(\alpha_j h)) - \nabla q_{t_n - \alpha_j h}(x_{t_n - \alpha_j h}) \right\|^2 + 3\mathbb{E} \left\| \nabla q_s(x_s) - \nabla q_s(x_n^*(t_n - s)) \right\|^2
$$

$$
+ 3\mathbb{E} \left\| \int_s^{t_n - \alpha_j h} \partial_u \nabla \ln q_u(x_u) du \right\|^2
$$

$$
\leq 3L^2 \cdot \mathbb{E} \left\| x_n^*(\alpha_j h) - x_{t_n - \alpha_j h} \right\|^2 + 3L^2 \cdot \mathbb{E} \left\| x_n^*(t_n - s) - x_s \right\|^2 + 3\mathbb{E} \left\| \int_s^{t_n - \alpha_j h} \partial_u \nabla \ln q_u(x_u) du \right\|^2.
$$

The second to third line is by Young's inequality, and the fourth to fifth line is by Theorem 2.2. By Lemma 3 in (Chen et al., 2023b),

$$\mathbb{E}\left\|\int_s^{t_n-\alpha_j h} \partial_u \nabla \ln q_u(x_u) du\right\|^2 \le \delta \cdot \int_s^{t_n-\alpha_j h} \mathbb{E}\left\|\partial_u \nabla \ln q_u(x_u)\right\|^2 du$$

$$\le \delta \cdot \int_s^{t_n-\alpha_j h} L^2 d \max(L, \frac{1}{u}) du \le L^2 d\delta^2(L \vee \frac{1}{t_n-h}).$$

Now, by Theorem B.1 and the fact that $h \lesssim \frac{1}{L}$,

$$\mathbb{E}\left\|\nabla q_{t_n-\alpha_j h}(x_n^*(\alpha_j h)) - \nabla \ln q_s(x_n^*(t_n-s))\right\|^2 \le 12L^2 \exp(Lh) \mathbb{E}\left\|\widehat{x}_n - x_{t_n}\right\|^2 + 3L^2 d\delta^2(L \vee \frac{1}{t_n-h})$$

$$\le 36L^2 \mathbb{E}\left\|\widehat{x}_n - x_{t_n}\right\|^2 + 3L^2 d\delta^2(L \vee \frac{1}{t_n-h}).$$

We conclude that the term in Equation (24) can be bounded by

$$R \cdot \delta \cdot e^{2\alpha_i h} \cdot \sum_{j=1}^i \int_{t_n-\min(j\delta,\alpha_i h)}^{t_n-(j-1)\delta} 36L^2 \mathbb{E}\left\|\widehat{x}_n - x_{t_n}\right\|^2 + 3L^2 d\delta^2(L \vee \frac{1}{t_n-h}) \, \mathrm{d}s$$

$$\le 2R^2 \delta^2 \cdot \left(36L^2 \mathbb{E}\left\|\widehat{x}_n - x_{t_n}\right\|^2 + 3L^2 d\delta^2(L \vee \frac{1}{t_n-h})\right)$$

$$= 4h^2 L^2 \left(18 \mathbb{E}\left\|\widehat{x}_n - x_{t_n}\right\|^2 + \frac{3}{2}d\delta^2(L \vee \frac{1}{t_n-h})\right).$$

Combining the bounds for Equation (23) and Equation (24), we get

$$\mathbb{E}\left\|\widehat{x}_{n,i}^{(k)} - x_n(\alpha_i h)\right\|^2 \lesssim h^2 L^2 \cdot \left(\frac{1}{R}\sum_{j=1}^R \left\|\widehat{x}_{n,j}^{(k-1)} - x_n^*(\alpha_j h)\right\|^2 + \frac{\epsilon_{\mathrm{sc}}^2}{L^2} + d\delta^2(L \vee \frac{1}{t_n-h}) + \mathbb{E}\left\|\widehat{x}_n - x_{t_n}\right\|^2\right).$$

Given sufficiently small constant $h$, $e^{\alpha_i h} \le 2$. Moreover, by the definition of $\delta$, $R\delta = h$. By unrolling the recursion, we get

$$\mathbb{E}\left\|\widehat{x}_{n,i}^{(k)} - x_{t_n-\alpha_i h}\right\|^2 \lesssim \left(8h^2 L^2\right)^k \cdot \left(\frac{1}{R}\sum_{j=1}^R \left\|x_{n,j}^{(0)} - x_n^*(\alpha_j h)\right\|^2\right)$$

$$+ \frac{8h^2 L^2}{1-8h^2 L^2}\left(\frac{\epsilon_{\mathrm{sc}}^2}{L^2} + d\delta^2(L \vee \frac{1}{t_n-h}) + \mathbb{E}\left\|\widehat{x}_n - x_{t_n}\right\|^2\right)$$

$$\lesssim \left(8h^2 L^2\right)^k \cdot \left(\frac{1}{R}\sum_{j=1}^R \left\|\widehat{x}_{n,j}^{(0)} - x_n^*(\alpha_j h)\right\|^2\right)$$

$$+ h^2 \left(\epsilon_{\mathrm{sc}}^2 + L^2 d\delta^2(L \vee \frac{1}{t_n-h}) + L^2 \cdot \mathbb{E}\left\|\widehat{x}_n - x_{t_n}\right\|^2\right). \qquad \square$$

As a consequence of Lemma C.2, if we take the number $K_n$ of Picard iterations sufficiently large, the error incurred in our estimate for $x_{t_n-\alpha_i h_n}$ is dominated by the terms in the second line of Eq. equation 22.

**Corollary C.3.** *Assume $L \ge 1$. For all $n \in \{0, \cdots, N-1\}$, suppose we draw $\widehat{x}_n$ from an arbitrary distribution $p_n$, then run step (a) - (e) in Algorithm 7. In addition, suppose $h_n < \frac{1}{3L}$ and $K_n \gtrsim \log(R_n)$. Then for any $i \in \{1, \cdots, R_n\}$,*

$$\mathbb{E}\left\|\widehat{x}_{n,i}^{(K_n)} - x_n^*(\alpha_i h_n)\right\|^2 \lesssim h_n^2 \epsilon_{\mathrm{sc}}^2 + \frac{L^2 d h_n^4}{R_n^2}\left(L \vee \frac{1}{t_n-h_n}\right) + L^2 h_n^2 \cdot \mathbb{E}\left\|\widehat{x}_n - x_{t_n}\right\|^2,$$

*where $x_n^*(t)$ is solution of the true ODE starting at $\widehat{x}_n$ at time $t_n$ and running until time $t_n - t$.*

*Proof.* Fixing iteration $n$, we will let $h := h_n$, $R := R_n$, $\delta := \delta_n$ and $K := K_n$. Notice that when $K \geq \frac{2}{\log \frac{1}{8h^2 L^2}} \cdot \log(R)$, $\left(8h^2 L^2\right)^K$ is at most $\frac{1}{R^2}$. Now by plugging Theorem C.1 into Theorem C.2, we get

$$
\mathbb{E} \left\| \widehat{x}_{n,i}^{(K)} - x_n^*(\alpha_i h) \right\|^2 \lesssim \left(8h^2 L^2\right)^K \cdot h^2 \cdot \left( \epsilon_{\mathrm{sc}}^2 + L^2 dh^2 (L \vee \frac{1}{t_n - h}) + L^2 \left\| \widehat{x}_n - x_{t_n} \right\|^2 \right)
$$

$$
+ h^2 \left( \epsilon_{\mathrm{sc}}^2 + L^2 d\delta^2 (L \vee \frac{1}{t_n - h}) + L^2 \cdot \mathbb{E} \left\| \widehat{x}_n - x_{t_n} \right\|^2 \right)
$$

$$
\lesssim \left( (8h^2 L^2)^K + 1 \right) h^2 \cdot \left( \epsilon_{\mathrm{sc}}^2 + L^2 \cdot \mathbb{E} \left\| \widehat{x}_n - x_{t_n} \right\|^2 \right)
$$

$$
+ \left( (8h^2 L^2)^K + \frac{1}{R^2} \right) \cdot L^2 dh^4 \left( L \vee \frac{1}{t_n - h} \right)
$$

$$
\lesssim h^2 \epsilon_{\mathrm{sc}}^2 + L^2 h^2 \cdot \mathbb{E} \left\| \widehat{x}_n - x_{t_n} \right\|^2 + \frac{L^2 dh^4}{R^2} \left( L \vee \frac{1}{t_n - h} \right)
$$

The first to second inequality by rearrangement of terms, while the second to third inequality is by the fact that the terms $\frac{1}{R^2}$ dominates $\left(8h^2 L^2\right)^K$. $\qquad \square$

We can now prove the parallel analogue of Theorem B.3 and Theorem B.4. Note that the bounds in Theorem C.4 and Theorem C.5 are identical to the bounds in Theorem B.3 and Theorem B.4, except from an additional $\frac{1}{R_n^2}$ factor for the middle term. This additional factor stems from using $R_n$ midpoints in each iteration $n$ (compared to using one midpoint each iteration in Algorithm 6).

**Lemma C.4** (Parallel Predictor Bias). *Assume $L \geq 1$. For all $n \in \{0, \cdots, N-1\}$, suppose we draw $\widehat{x}_n$ from an arbitrary distribution $p_n$, then run step (a) - (e) in Algorithm 7. In addition, suppose $h_n < \frac{1}{3L}$ and $K_n \gtrsim \log(R_n)$. Then we have*

$$
\mathbb{E} \left\| \mathbb{E}_\alpha \widehat{x}_{n+1} - x_n^*(h_n) \right\|^2 \lesssim h_n^2 \cdot \epsilon_{\mathrm{sc}}^2 + \frac{L^4 h_n^6 d}{R_n^2} \cdot (L \vee \frac{1}{t_n - h_n}))) + L^2 h_n^2 \cdot \mathbb{E} \left\| \widehat{x}_n - x_{t_n} \right\|^2,
$$

*where $x_n^*(t)$ is solution of the true ODE starting at $\widehat{x}_n$ at time $t_n$ and running until time $t_n - t$.*

*Proof.* Fixing iteration $n$, we will let $h := h_n$, $R := R_n$, $\delta := \delta_n$ and $K := K_n$. We have

$$
\mathbb{E} \left\| \mathbb{E}_\alpha \widehat{x}_{n+1} - x_n^*(h) \right\|^2
$$

$$
\leq \mathbb{E} \left\| \mathbb{E}_\alpha \left[ \delta \cdot \sum_{i=1}^R e^{h - \alpha_i h} \cdot \widehat{s}_{t_n - \alpha_i h}(\widehat{x}_{n,i}^{(K)}) \right] - \int_{t_n - h}^{t_n} e^{s - (t_n - h)} \nabla \ln q_s(x_n^*(t_n - s)) \, \mathrm{d}s \right\|^2
$$

$$
\leq 2 \mathbb{E} \left\| \mathbb{E}_\alpha \left[ \sum_{i=1}^R \delta e^{h - \alpha_i h} \cdot \left( \widehat{s}_{t_n - \alpha_i h}(\widehat{x}_{n,i}^{(K)}) - \nabla \ln q_{t_n - \alpha_i h}(x_n^*(\alpha_i h)) \right) \right] \right\|^2 \tag{26}
$$

$$
+ 2 \mathbb{E} \left\| \mathbb{E}_\alpha \sum_{i=1}^R \delta e^{h - \alpha_i h} \cdot \nabla \ln q_{t_n - \alpha_i h}(x_n^*(\alpha_i h)) - \int_{t_n - h}^{t_n} e^{s - (t_n - h)} \nabla \ln q_s(x_n^*(t_n - s)) \, \mathrm{d}s \right\|^2. \tag{27}
$$

Since $\alpha_i$ is drawn uniformly from $[(i-1)\delta, i\delta]$,

$$
\mathbb{E}_\alpha \left[ \delta e^{h - \alpha_i h} \cdot \nabla \ln q_{t_n - \alpha_i h}(x_n^*(\alpha_i h)) \right] = \int_{t_n - i\delta}^{t_n - (i-1)\delta} e^{s - (t_n - h)} \nabla \ln q_s(x_n^*(t_n - s)) \, \mathrm{d}s,
$$

and the term in Equation (27) is equal to 0. Now we need to bound Equation (26).

$$\mathbb{E}\left\|\widehat{s}_{t_n-\alpha_i h}(\widehat{x}_{n,i}^{(K)}) - \nabla\ln q_{t_n-\alpha_i h}(x_n^*(\alpha_i h))\right\|^2$$

$$\lesssim \mathbb{E}\left\|\widehat{s}_{t_n-\alpha_i h}(\widehat{x}_{n,i}^{(K)}) - \widehat{s}_{t_n-\alpha_i h}(x_{t_n-\alpha_i h})\right\|^2 + \mathbb{E}\left\|\nabla\ln q_{t_n-\alpha_i h}(x_{t_n-\alpha_i h}) - \nabla\ln q_{t_n-\alpha_i h}(x_n^*(\alpha_i h))\right\|^2$$

$$+ \mathbb{E}\left\|\widehat{s}_{t_n-\alpha_i h}(x_{t_n-\alpha_i h}) - \nabla\ln q_{t_n-\alpha_i h}(x_{t_n-\alpha_i h})\right\|^2$$

$$\lesssim L^2\cdot\mathbb{E}\left\|\widehat{x}_{n,j}^{(K)} - x_n^*(\alpha_j h)\right\|^2 + L^2\cdot\mathbb{E}\left\|x_n^*(\alpha_j h) - x_{t_n-\alpha_i h}\right\|^2 + \epsilon_{\mathrm{sc}}^2$$

$$\lesssim L^2\cdot\left(h^2\epsilon_{\mathrm{sc}}^2 + \frac{L^2 dh^4}{R^2}\left(L\vee\frac{1}{t_n-h}\right) + L^2 h^2\cdot\mathbb{E}\left\|\widehat{x}_n - x_{t_n}\right\|^2\right) + L^2\,\mathbb{E}\left\|\widehat{x}_n - x_{t_n}\right\|^2 + \epsilon_{\mathrm{sc}}^2.$$

The first to second line by inequality $(\sum_{i=1}^n a_i)^2 \le n\sum_{i=1}^n a_i^2$, the second to third line is by Theorem 2.4, Theorem 2.2 and Theorem 2.3, and the third to fourth line is by Theorem C.3 ($K\gtrsim \log(R)$, which satisfies the condition in Theorem C.3) and Theorem B.1. Hence

$$2\,\mathbb{E}\left\|\mathbb{E}_\alpha\left[\sum_{i=1}^R \delta e^{h-\alpha_i h}\cdot\left(\widehat{s}_{t_n-\alpha_i h}(\widehat{x}_{n,i}^{(K)}) - \nabla\ln q_{t_n-\alpha_i h}(x_{t_n-\alpha_i h})\right)\right]\right\|^2$$

$$\le 2R(2\delta)^2\sum_{i=1}^R\mathbb{E}_\alpha\left\|\widehat{s}_{t_n-\alpha_i h}(\widehat{x}_{n,i}^{(K)}) - \nabla\ln q_{t_n-\alpha_i h}(x_{t_n-\alpha_i h})\right\|^2$$

$$\lesssim 8\delta^2 R^2\cdot\left(\epsilon_{\mathrm{sc}}^2 + L^2\cdot\left(h^2\epsilon_{\mathrm{sc}}^2 + \frac{L^2 dh^4}{R^2}\left(L\vee\frac{1}{t_n-h}\right) + L^2 h^2\cdot\mathbb{E}\left\|\widehat{x}_n - x_{t_n}\right\|^2\right) + L^2\,\mathbb{E}\left\|\widehat{x}_n - x_{t_n}\right\|^2\right)$$

$$\lesssim h^2\cdot\epsilon_{\mathrm{sc}}^2 + \frac{L^4 dh^6}{R^2}(L\vee\frac{1}{t_n-h}) + L^2 h^2\,\mathbb{E}\left\|\widehat{x}_n - x_{t_n}\right\|^2$$

The first to second step is by inequality $(\sum_{i=1}^n a_i)^2 \le n\sum_{i=1}^n a_i^2$ and Young's inequality, the second to third line is by plugging in our previous calculation, and the third to forth line is by $h = \delta R$ and that $h\lesssim 1/L$. $\qquad\square$

**Lemma C.5** (Parallel Predictor Variance). *Assume $L\ge 1$. For all $n\in\{0,\cdots,N-1\}$, suppose we draw $\widehat{x}_n$ from an arbitrary distribution $p_n$, then run step (a) - (e) in Algorithm 7. In addition, suppose $h_n < \frac{1}{3L}$ and $K_n\gtrsim\log(R_n)$. Then we have*

$$\mathbb{E}_\alpha\left\|\widehat{x}_{n+1} - x_n^*(h_n)\right\|^2 \lesssim h_n^2\cdot\epsilon_{\mathrm{sc}}^2 + \frac{L^2 dh_n^4}{R_n^2}\left(L\vee\frac{1}{t_n-h_n}\right) + L^2 h_n^2\cdot\mathbb{E}\left\|\widehat{x}_n - x_{t_n}\right\|^2,$$

*where $x_n^*(t)$ is solution of the true ODE starting at $\widehat{x}_n$ at time $t_n$ and running until time $t_n - t$.*

*Proof.* Fixing iteration $n$, we will let $h := h_n$, $R := R_n$, $\delta := \delta_n$ and $K := K_n$. We will separate $\mathbb{E}_\alpha\left\|\widehat{x}_{n+1} - x_n^*(h)\right\|^2$ into several terms and bound each term separately.

$$\mathbb{E}_\alpha\left\|\widehat{x}_{n+1} - x_n^*(h)\right\|^2$$

$$\le \mathbb{E}_\alpha\left\|\delta\cdot\sum_{i=1}^R e^{h-\alpha_i h}\cdot\widehat{s}_{t_n-\alpha_i h}(\widehat{x}_{n,i}^{(K)}) - \int_{t_n-h}^{t_n} e^{s-(t_n-h)}\nabla\ln q_s(x_n^*(t_n-s))\,\mathrm{d}s\right\|^2$$

$$\le 3\,\mathbb{E}_\alpha\left\|\sum_{i=1}^R \delta e^{h-\alpha_i h}\cdot\left(\widehat{s}_{t_n-\alpha_i h}(\widehat{x}_{n,i}^{(K)}) - \nabla\ln q_{t_n-\alpha_i h}(x_{t_n-\alpha_i h})\right)\right\|^2 \tag{28}$$

$$+ 3\,\mathbb{E}\left\|\sum_{i=1}^R\int_{t_n-i\delta}^{t_n-(i-1)\delta} e^{h-\alpha_i h}\cdot\left(\nabla\ln q_{t_n-\alpha_i h}(x_{t_n-\alpha_i h}) - \nabla\ln q_s(x_n^*(t_n-s))\right)\,\mathrm{d}s\right\|^2 \tag{29}$$

$$+ 3\,\mathbb{E}\left\|\sum_{i=1}^R\int_{t_n-i\delta}^{t_n-(i-1)\delta}\left(e^{h-\alpha_i h} - e^{s-(t_n-h)}\right)\cdot\nabla\ln q_s(x_n^*(t_n-s))\,\mathrm{d}s\right\|^2. \tag{30}$$

Equation (28) is identical to Equation (26) in Theorem C.4, and can be bounded by

$$Equation \ (28) \lesssim h^2 \cdot \epsilon_{\text{sc}}^2 + L^2 h^2 \cdot \mathbb{E} \left\| \widehat{x}_n - x_{t_n} \right\|^2 + \frac{L^4 d h^6}{R^2} (L \vee \frac{1}{t_n - h}))).$$

Next we will bound Equation (29). By Theorem E.1 and Theorem B.1,

$$\mathbb{E} \left\| \nabla q_{t_n - \alpha_j h}(x_n^*(\alpha_j h)) - \nabla \ln q_s(x_n^*(t_n - s)) \right\|^2 \lesssim L^2 d \delta^2 (L \vee \frac{1}{t_n - h}) + L^2 \cdot \mathbb{E} \left\| \widehat{x}_n - x_{t_n} \right\|^2,$$

hence Equation (29) can be bounded with similar calculations as for Equation (24) in Theorem C.2, by the following term:

$$12 R^2 \delta^2 \cdot \left( L^2 d \delta^2 (L \vee \frac{1}{t_n - h}) + O(L^2) \cdot \mathbb{E} \left\| \widehat{x}_n - x_{t_n} \right\|^2 \right) \lesssim \frac{L^2 d h^4}{R^2} (L \vee \frac{1}{t_n - h}) + L^2 h^2 \, \mathbb{E} \left\| \widehat{x}_n - x_{t_n} \right\|^2.$$

Finally we will bound Equation (30) Since both $\alpha_i$ and $s$ belong to the range $[(i-1)\delta, i\delta]$,

$$e^{h - \alpha_i h} - e^{s - (t_n - h)} \leq e^h (e^{-(i-1)\delta} - e^{-i\delta}) \leq e^h \cdot \delta \leq 2\delta.$$

Moreover, by the fact that $\mathbb{E} \left\| \nabla \ln q_s(x_{t_n}) \right\|^2 \leq Ld$ (by integration by parts), Theorem E.1, Theorem B.1 and the fact that $L\delta = o(1)$, we have

$$\mathbb{E} \left\| \nabla \ln q_s(x_n^*(t_n - s)) \right\|^2 \lesssim \mathbb{E} \left\| \nabla \ln q_s(x_{t_n}) \right\|^2 + \mathbb{E} \left\| \nabla \ln q_s(x_n^*(t_n - s)) - \nabla \ln q_s(x_{t_n}) \right\|^2$$
$$\lesssim Ld + L^2 \exp(L\delta) \left\| \widehat{x}_n - x_{t_n} \right\|^2 \lesssim Ld + L^2 \left\| \widehat{x}_n - x_{t_n} \right\|^2.$$

Hence Equation (30) can be bounded by

$$3 \mathbb{E} \left\| \sum_{i=1}^R \int_{t_n - i\delta}^{t_n - (i-1)\delta} \left( e^{h - \alpha_i h} - e^{s - (t_n - h)} \right) \cdot \nabla \ln q_s(x_n^*(t_n - s)) \, \mathrm{d}s \right\|^2$$
$$\leq 3 R \delta \sum_{i=1}^R \int_{t_n - i\delta}^{t_n - (i-1)\delta} \mathbb{E} \left\| 2\delta \nabla \ln q_s(x_n^*(t_n - s)) \right\|^2$$
$$\lesssim 3 R \delta \cdot 4 \delta^2 \sum_{i=1}^R \int_{t_n - i\delta}^{t_n - (i-1)\delta} \left( Ld + L^2 \left\| \widehat{x}_n - x_{t_n} \right\|^2 \right)$$
$$\lesssim R^2 \delta^4 \left( Ld + L^2 \left\| \widehat{x}_n - x_{t_n} \right\|^2 \right) = \frac{L d h^4}{R^2} + \frac{L^2 h^4}{R^2} \left\| \widehat{x}_n - x_{t_n} \right\|^2.$$

By adding together Equation (28), Equation (29) and Equation (30), and combining terms, we conclude that

$$\mathbb{E}_{\alpha} \left\| \widehat{x}_{n+1} - x_n^*(h) \right\|^2 \lesssim h^2 \cdot \epsilon_{\text{sc}}^2 + L^2 h^2 \cdot \mathbb{E} \left\| \widehat{x}_n - x_{t_n} \right\|^2 + \frac{L^4 d h^6}{R^2} (L \vee \frac{1}{t_n - h})))$$
$$+ \frac{L^2 d h^4}{R^2} (L \vee \frac{1}{t_n - h}) + L^2 h^2 \, \mathbb{E} \left\| \widehat{x}_n - x_{t_n} \right\|^2 + \frac{L d h^4}{R^2} + \frac{L^2 h^4}{R^2} \left\| \widehat{x}_n - x_{t_n} \right\|^2$$
$$\lesssim h^2 \cdot \epsilon_{\text{sc}}^2 + \frac{L^2 d h^4}{R^2} (L \vee \frac{1}{t_n - h}) + L^2 h^2 \, \mathbb{E} \left\| \widehat{x}_n - x_{t_n} \right\|^2. \qquad \square$$

We can now prove our main guarantee for the parallel predictor step, which states that with logarithmically many parallel rounds and $\widetilde{O}(\sqrt{d})$ score estimate queries over a short time interval (of length $O(1/L)$) of the reverse process, the algorithm does not drift too far from the true ODE. Our proof follows a similar flow as Theorem B.5.

Note that due to the existence of $R_n$ midpoints in each step, we can set $h_n = \Theta(1)$. We will set $h_n$ to be $\Theta(\frac{1}{L})$, unless $t_n$ is close to the end time $\delta$ (see Algorithm 9 for the global algorithm and timeline). If $t_n$ is close to $\delta$, we will repeated half $h_n$ as $n$ increases, until we reach the end time.

**Theorem C.6.** *Assume $L \geq 1$. Let $\beta \geq 1$ be an adjustable parameter. When we set $h_n = \min\{\frac{1}{4L}, t_n/2, t_n - \delta\}$, $K \gtrsim \log(\frac{\beta\sqrt{d}}{\epsilon})$ and $R_n = h_n \cdot \beta \cdot \frac{L\sqrt{d}}{\epsilon} = O\left(\frac{\beta\sqrt{d}}{\epsilon}\right)$, the Wasserstein distance between the true ODE process and the process in Algorithm 7, both starting from $\hat{x}_0 \sim q_{t_0}$ and run for total time $T_N \lesssim \frac{1}{L}$ is bounded by*

$$W_2(\hat{x}_N, x_{T_N}) \lesssim \frac{\epsilon_{\mathrm{sc}}}{L} + \frac{\epsilon}{\beta\sqrt{L}}.$$

*Proof.* To avoid confusion, we will reserve $\hat{x}_n$ as the result of running Algorithm 7 for $n$ steps, starting at $\hat{x}_0 \sim q_0$. We will use $\hat{y}_n$ to denote the result from running the true ODE process, starting at $\hat{x}_{n-1}$. Then by an identical calculation as in Theorem B.5,

$$\mathbb{E}_\alpha \left\| x_{t_N} - \hat{x}_N \right\|^2 \leq \exp\left(O(Lh_{N-1})\right) \left\| x_{t_{N-1}} - \hat{x}_{N-1} \right\|^2 + \frac{2}{Lh_{N-1}} \left\| \mathbb{E}_\alpha \hat{x}_N - y_N \right\|^2 + \mathbb{E}_\alpha \left\| \hat{x}_N - y_N \right\|^2.$$

Now we do a similar calculation as in Theorem B.5, but utilizes the bias and variance of one step in the parallel algorithm instead of sequential. Taking the expectation wrt $\hat{x}_0 \sim q_{t_0}$, by Lemmas C.4 and C.5,

$$\mathbb{E}\left\| x_{t_N} - \hat{x}_N \right\|^2$$

$$\leq \exp\left(O(Lh_{N-1})\right) \mathbb{E}\left\| x_{t_{N-1}} - \hat{x}_{N-1} \right\|^2 + \frac{2}{Lh_{N-1}} \mathbb{E}\left\| \mathbb{E}_\alpha \hat{x}_N - y_N \right\|^2 + \mathbb{E}\left\| \hat{x}_N - y_N \right\|^2$$

$$\leq \exp\left(O(Lh_{N-1})\right) \mathbb{E}\left\| x_{t_{N-1}} - \hat{x}_{N-1} \right\|^2$$
$$+ O\left( \frac{1}{Lh_{N-1}} \left( h_{N-1}^2 \epsilon_{\mathrm{sc}}^2 + \frac{L^4 dh_{N-1}^6}{R_n^2} \left( L \vee \frac{1}{t_{N-1}} \right) + L^2 h_{N-1}^2 \mathbb{E}\left\| x_{t_{N-1}} - \hat{x}_{N-1} \right\|^2 \right) \right)$$
$$+ O\left( h_{N-1}^2 \epsilon_{\mathrm{sc}}^2 + \frac{L^2 dh_{N-1}^4}{R_n^2} \left( L \vee \frac{1}{t_{N-1}} \right) + L^2 h_{N-1}^2 \mathbb{E}\left\| x_{t_{N-1}} - \hat{x}_{N-1} \right\|^2 \right)$$

$$\leq \exp\left(O(Lh_{N-1})\right) \mathbb{E}\left\| x_{t_{N-1}} - \hat{x}_{N-1} \right\|^2$$
$$+ O\left( \frac{h_{N-1}\epsilon_{\mathrm{sc}}^2}{L} + h_{N-1}^2 \epsilon_{\mathrm{sc}}^2 + \frac{L^3 dh_{N-1}^5 + L^2 dh_{N-1}^4}{R_n^2} \left( L \vee \frac{1}{t_{N-1}} \right) \right)$$

Since $h_{N-1} < \frac{1}{L}$, the term $L^3 dh_{N-1}^5$ is dominated by the term $L^2 dh_{N-1}^4$ and the term $h_{N-1}^2 \epsilon_{\mathrm{sc}}^2$ os dominated by $\frac{h_{N-1}\epsilon_{\mathrm{sc}}^2}{L}$. By induction, noting that $x_{t_0} = \hat{x}_0$, we have

$$\mathbb{E}\left\| x_{t_N} - \hat{x}_N \right\|^2 \lesssim \sum_{n=0}^{N-1} \left( \frac{h_n}{L}\epsilon_{\mathrm{sc}}^2 + \frac{L^2 dh_n^4}{R_n^2} \cdot \left( L \vee \frac{1}{t_n} \right) \right) \cdot \exp\left( O\left( L \sum_{i=n+1}^{N-1} h_i \right) \right).$$

Since $\sum_{i=n+1}^{N-1} h_i \leq \frac{1}{L}$, $\exp\left( O\left( L \sum_{i=n+1}^{N-1} h_i \right) \right)$ is a constant. Moreover, by our choice of $h_n$, it is always the case that $h_n \leq \frac{t_n}{2} \leq t_n - h_n$ and that $h_n \leq \frac{1}{L}$. Therefore

$$\frac{L^2 dh_n^4}{R_n^2} \left( L \vee \frac{1}{t_n - h_n} \right) \leq \frac{L^2 dh_n^3}{R_n^2} \leq \frac{L^2 dh_n}{\frac{\beta^2 L^2 d}{\epsilon^2}} = \frac{h_n \epsilon^2}{\beta^2},$$

and thus

$$\mathbb{E}\left\| x_{t_N} - \hat{x}_N \right\|^2 \lesssim \sum_{n=0}^{N-1} \frac{h_n}{L}\epsilon_{\mathrm{sc}}^2 + \frac{h_n \epsilon^2}{\beta^2} \lesssim \frac{\epsilon_{\mathrm{sc}}^2}{L^2} + \frac{\epsilon^2}{L\beta^2}.$$

We conclude that

$$W_2(x_{t_N}, \hat{x}_N) = \sqrt{\mathbb{E}\left\| x_{t_N} - \hat{x}_N \right\|^2} \lesssim \frac{\epsilon_{\mathrm{sc}}}{L} + \frac{\epsilon}{\beta\sqrt{L}}. \qquad \square$$

## C.2 Corrector step

In this step we will be using the parallel algorithm in (Anari et al., 2024) to estimate the underdamped Langevin diffusion process. Since we will be fixing the score function in time, we will use $\nabla \ln q$ to

denote the true score function for the diffusion process, and $\widehat{s}$ to denote the estimated score function. We will choose the friction parameter to be $\gamma \asymp \sqrt{L}$.

---

**Algorithm 8** CORRECTOR STEP (PARALLEL) (Anari et al., 2024)

---

**Input parameters:**

- Starting sample $(\widehat{x}_0, \widehat{v}_0) \sim p \otimes \mathcal{N}(0, I_d)$, Number of steps $N$, Step size $h$, Score estimates $\widehat{s} \approx \nabla \ln q$, Number of midpoint estimates $R$, $\delta := \frac{h}{R}$

1. For $n = 0, \ldots, N - 1$:

   (a) Let $t_n = nh$

   (b) Let $(\widehat{x}_{n,i}^{(k)}, \widehat{v}_{n,i}^{(k)})$ represent the algorithmic estimate of $(x_{t_n + ih/R}, v_{t_n + ih/R})$ at iteration $k$.

   (c) Let $(\zeta^x, \zeta^v)$ be a correlated gaussian vector corresponding to change caused by the Brownian motion term in $h/R$ time (see more detail in (Anari et al., 2024))

   (d) For $i = 0, \cdots, R$ in parallel: Let $(\widehat{x}_{n,i}^{(0)}, \widehat{v}_{n,i}^{(0)}) = (\widehat{x}_n, \widehat{v}_n)$

   (e) For $k = 1, \cdots, K$:

   For $i = 1, \cdots, R$ in parallel:
   $$\widehat{x}_{n,i}^{(k)} := \widehat{x}_{n,i-1}^{(k-1)} + \frac{1 - \exp(-\gamma h/R)}{\gamma} \cdot \widehat{v}_{n,i-1}^{(k-1)} - \frac{h/R - (1 - \exp(-\gamma h/R))/\gamma}{\gamma} \cdot \widehat{s}(x_{n,j}^{k-1}) + \zeta^x$$
   $$\widehat{v}_{n,i}^{(k)} = \exp(-\gamma h/R) \cdot \widehat{v}_{n,i-1}^{(k-1)} - \frac{1 - \exp(-\gamma h/R)}{\gamma} \cdot \widehat{s}(x_{n,i-1}^{(k-1)}) + \zeta^v$$

   (f) $(\widehat{x}_{n+1}, \widehat{v}_{n+1}) = (\widehat{x}_{n,R}^K, \widehat{v}_{n,R}^K)$

2. Let $t_N = Nh$

3. Return $\widehat{x}_N, t_N$.

---

Let $T_N$ denote the total time the parallel corrector step is run (namely, $T_N = nh$). Consider two continuous underdamped Langevin diffusion processes $u^*(t) = (x^*(t), v^*(t))$ and $u_{t_0+t} = (x_{t_0+t}, v_{t_0+t})$ with coupled brownian motions. The first one start from position $x^*(0) = \widehat{x}_0 \sim p$ and the second one start from position $x_{t_0} \sim q$. Both processes start with velocity $v^*(0) = v_{t_0} \sim \mathcal{N}(0, I_d)$. We will bound both the distance measure between $x^*(t)$ and the true sample $x_{t_0+t}$, and the distance measure between $x^*(t_N)$ and outputs of Algorithm 8. First, (Chen et al., 2023b) gives the following bound on the total variation error between $x^*(T_N)$ and $x_{t_N}$.

**Lemma C.7** ((Chen et al., 2023b), Lemma 9). *If $h \lesssim \frac{1}{\sqrt{L}}$, then*

$$\mathsf{TV}(x^*(T_N), x_{t_N}) \lesssim \frac{W_2(p, q)}{L^{1/4} T_N^{3/2}}.$$

Next, (Anari et al., 2024) bounds the discretization error in Algorithm 8 in terms of quantities that relates to the supremum of $\mathbb{E} \|\nabla \ln q(x^*(t))\|^2$ and $\mathbb{E} \|v^*(t)\|^2$ where $t \in [0, T_N]$.

**Lemma C.8** ((Anari et al., 2024), Theorem 20, Implicit). *Assume $L \geq 1$. In Algorithm 8, assume $K \gtrsim \log(d)$ (for sufficiently large constant), $K \lesssim 4 \log R$ and $h \lesssim \frac{1}{\sqrt{L}}$. Then*

$$\mathsf{KL}(\widehat{x}_N, x^*(T_N)) \lesssim \frac{T_N}{\sqrt{L}} \cdot \left( \epsilon_{\mathrm{sc}}^2 + L^2 (\frac{\gamma d h^3}{R^4} + \frac{h^2}{R^2} \mathcal{P} + \frac{h^4}{R^4} \mathcal{Q}) \right),$$

*where $\mathcal{P} = \sup_{t \in [0, T_N]} \mathbb{E}[\|v^*(t)\|^2]$ and $\mathcal{Q} = \sup_{t \in [0, T_N]} \mathbb{E}[\|\nabla \ln q(x^*(t))\|^2]$.*

To reason about the value of $\mathcal{P}$ and $\mathcal{Q}$, we will use the following lemma in (Chen et al., 2023b).

**Lemma C.9** ((Chen et al., 2023b), Lemma 10). *For any $t \lesssim \frac{1}{\sqrt{L}}$,*

$$\mathbb{E} \|u^*(t) - u_{t_0+t}\|^2 \lesssim W_2^2(p, q).$$

**Lemma C.10.** *Assume $L \geq 1$. For any $T_N \lesssim \frac{1}{\sqrt{L}}$,*

$$\sup_{t \in [0, T_N]} \mathbb{E}[\|\nabla \ln q(x^*(t))\|^2] \lesssim L^2 W_2^2(p, q) + Ld$$

*and*

$$\sup_{t \in [0, T_N]} \mathbb{E}\|v^*(t)\|^2 \lesssim W_2^2(p, q) + d.$$

*Proof.* Note that $(q, \mathcal{N}(0, I_d))$ is a stationary distribution of the underdamped Langevin diffusion process, hence $x_t \sim q$ and $v_t \sim \mathcal{N}(0, I_d)$. Hence $\mathbb{E}\|\nabla \ln q(x_t)\|^2 \leq Ld$ by integration by parts. Similarly, $\mathbb{E}\|v_t\|^2 = \mathbb{E}[\|\mathcal{N}(0, I_d)\|^2] \lesssim d$. Since $T_N \lesssim \frac{1}{\sqrt{L}}$, for any $t \in [0, T_N]$, we can now bound $\mathbb{E}\|\nabla \ln q(x^*(t))\|^2$ and $\mathbb{E}\|v^*(t)\|^2$ by Theorem C.9 as follows:

$$\mathbb{E}\|\nabla \ln q(x^*(t))\|^2 \leq 2 \cdot \mathbb{E}\|\nabla \ln q(x_t)\|^2 + 2 \cdot \mathbb{E}\|\nabla \ln q(x^*(t)) - \nabla \ln q(x_t)\|^2$$
$$\lesssim Ld + L^2 \mathbb{E}\|x^*(t) - x_t\|^2 \lesssim Ld + L^2 \mathbb{E}\|u^*(t) - u_t\|^2 \lesssim Ld + L^2 W_2^2(p, q),$$

and

$$\mathbb{E}\|v^*(t)\|^2 \leq 2 \cdot \mathbb{E}\|v_t\|^2 + 2 \cdot \mathbb{E}\|v^*(t) - v_t\|^2$$
$$\lesssim \mathbb{E}\|v_t\|^2 + \mathbb{E}\|u^*(t) - u_t\|^2 \lesssim d + W_2^2(p, q). \qquad \square$$

**Theorem C.11.** *Let $\beta \geq 1$ be an adjustable parameter. Algorithm 8 with parameter $h = \frac{1}{\sqrt{8L}}, R = \beta \cdot \Theta(\frac{\sqrt{d}}{\epsilon}), K = 4 \cdot \log(R)$ and $T_N \lesssim \frac{1}{\sqrt{L}}$ has discretization error*

$$\mathsf{TV}(\widehat{x}_N, x^*(T_N)) \lesssim \sqrt{\mathsf{KL}(\widehat{x}_N, x^*(T_N))} \lesssim \frac{\epsilon_{\mathrm{sc}}}{\sqrt{L}} + \frac{\epsilon}{\beta} + \frac{\epsilon}{\beta\sqrt{d}} \cdot W_2(p, q).$$

*Proof.* Since $T_N \lesssim \frac{1}{\sqrt{L}}$ and $h = \Theta(\frac{1}{\sqrt{L}})$, $N = O(1)$. Plugging Theorem C.10 into Theorem C.8, we get that

$$\mathsf{KL}(\widehat{x}_N, x^*(T_N)) \lesssim \frac{T_N}{\sqrt{L}} \cdot \left(\epsilon_{\mathrm{sc}}^2 + L^2 \left(\frac{\gamma d h^3}{R^4} + \frac{h^2}{R^2} \cdot (d + W_2^2(p, q)) + \frac{h^4}{R^4} \cdot (L^2 W_2^2(p, q) + Ld)\right)\right)$$
$$\lesssim \frac{1}{L} \cdot \left(\epsilon_{\mathrm{sc}}^2 + \frac{d}{R^4} + \frac{Ld}{R^2} + \frac{Ld}{R^4} + \left(\frac{L}{R^2} + \frac{L^2}{R^4}\right) \cdot W_2^2(p, q)\right)$$
$$= \frac{\epsilon_{\mathrm{sc}}^2}{L} + \frac{\epsilon^2}{\beta^2} + \frac{\epsilon^2}{\beta^2 d} W_2^2(p, q).$$

The first to second line is by combining terms and setting $h = \Theta(\frac{1}{\sqrt{L}}), \gamma = \Theta(\sqrt{L})$, and the second to third line is by setting $R = \beta \cdot \Theta(\frac{\sqrt{d}}{\epsilon})$. Taking the square root of $\mathsf{KL}(\widehat{x}_N, x^*(T_N))$ yields the claim. $\qquad \square$

**Theorem C.12.** *Let $\beta \geq 1$ be an adjustable parameter. When Algorithm 8 is initialized at $(\widehat{x}_0, \widehat{v}_0) \sim p \otimes \mathcal{N}(0, I_d)$, there exists parameters $h = \frac{1}{\sqrt{8L}}, R = \beta \cdot \Theta(\frac{\sqrt{d}}{\epsilon}), K = \Theta(\log(\frac{\beta^2 d}{\epsilon^2}))$ and $T_N \lesssim \frac{1}{\sqrt{L}}$ such that the total variation distance between the final output of Algorithm 8 and the true distribution can be bounded as*

$$\mathsf{TV}(\widehat{x}_N, x_{t_N}) \lesssim \frac{\epsilon_{\mathrm{sc}}}{\sqrt{L}} + \frac{\epsilon}{\beta} + \sqrt{L} \cdot W_2(p, q).$$

*Proof.* By triangle inequality, $\mathsf{TV}(\widehat{x}_N, x_{t_N}) \leq \mathsf{TV}(\widehat{x}_N, x^*(T_N)) + \mathsf{TV}(x^*(T_N), x_{t_N})$. Combining Theorem C.7 and Theorem C.11 yields the claim. $\qquad \square$

## C.3 END-TO-END ANALYSIS

---

**Algorithm 9** PARALLELALGORITHM

---

**Input parameters:**

- Start time $T$, End time $\delta$, Corrector steps time $T_{\text{corr}} \lesssim 1/\sqrt{L}$, Number of predictor-corrector steps $N_0$, Score estimates $\widehat{s}_t$

1. Draw $\widehat{x}_0 \sim \mathcal{N}(0, I_d)$.
2. For $n = 0, \ldots, N_0$:
   (a) Starting from $\widehat{x}_n$, run Algorithm 7 with starting time $T - n/L$ with total time $\min(1/L, T - n/L - \delta)$. Let the result be $\widehat{x}'_{n+1}$.
   (b) Starting from $\widehat{x}'_{n+1}$, run Algorithm 8 for total time $T_{\text{corr}}$ and score estimate $\widehat{s}_{T-(n+1)/L}$ to obtain $\widehat{x}_{n+1}$.
3. Return $\widehat{x}_{N_0+1}$.

---

**Theorem C.13** (Parallel End to End Error). *By setting $T = \Theta\left(\log\left(\frac{d \vee \mathrm{m}_2^2}{\epsilon^2}\right)\right)$, $T_{corr} = \frac{1}{\sqrt{L}}$, $\delta = \Theta\left(\frac{\epsilon^2}{L^2(d \vee \mathrm{m}_2^2)}\right)$, and $\beta_1 = \beta_2 = \Theta\left(L \log\left(\frac{d \vee \mathrm{m}_2^2}{\epsilon^2}\right)\right)$ in Algorithm 7 and Algorithm 8, when $\epsilon_{\text{sc}} \lesssim \widetilde{\Theta}(\frac{\epsilon}{\sqrt{L}})$, the total variation distance between the output of Algorithm 9 and the target distribution $x_0 \sim q^*$ is*

$$\mathsf{TV}(\widehat{x}_{N_0+1}, x_0) \lesssim \epsilon,$$

*with iteration complexity $\widetilde{\Theta}(L \cdot \log^2\left(\frac{Ld \vee \mathrm{m}_2^2}{\epsilon}\right))$.*

*Proof.* Let $x_{t_n}$ be the result of running the true ODE for time $T - t_n$, starting from $x_T \sim q^*$. Let $y'_n$ be the result of running the predictor step in step $n - 1$ of Algorithm 9, starting from $x_{t_{n-1}} \sim q_{t_{n-1}}$ and start time $t_{n-1}$. In addition, let $\widehat{y}_n$ be the result of the corrector step in step $n - 1$ of Algorithm 9, starting from $y'_n$.

We will first bound the error in one predictor + corrector step that starts at $t_{n-1} = T - (n-1)/L$. By triangle inequality of TV distance and data processing inequality (applied to $\widehat{x}_n$ and $\widehat{y}_n$),

$$\begin{aligned}
\mathsf{TV}(\widehat{x}_n, x_{t_n}) &\leq \mathsf{TV}(\widehat{x}_n, \widehat{y}_n) + \mathsf{TV}(\widehat{y}_n, x_{t_n}) \\
&\leq \mathsf{TV}(\widehat{x}_{n-1}, x_{t_{n-1}}) + \mathsf{TV}(\widehat{y}_n, x_{t_n})
\end{aligned} \tag{31}$$

By Theorem C.6 parametrized by $\beta_1$ and Theorem C.12 parametrized by $\beta_2$,

$$\begin{aligned}
\mathsf{TV}(\widehat{y}_n, x_{t_n}) &\lesssim \frac{\epsilon_{\text{sc}}}{\sqrt{L}} + \frac{\epsilon}{\beta_2} + \sqrt{L} \cdot W_2(y'_n, x_{t_n}) \\
&\lesssim \frac{\epsilon_{\text{sc}}}{\sqrt{L}} + \frac{\epsilon}{\beta_2} + \sqrt{L}\left(\frac{\epsilon_{\text{sc}}}{L} + \frac{\epsilon}{\beta_1 \sqrt{L}}\right) \\
&\lesssim \frac{\epsilon_{\text{sc}}}{\sqrt{L}} + \frac{\epsilon}{\min(\beta_1, \beta_2)}.
\end{aligned}$$

The first line is by Theorem C.12, and the first to second line is by Theorem C.12. Next, note that at the beginning of the process, $t_0 = T$, and at the end of the process, $t_{N_0+1} = \delta$. By induction on Equation (31),

$$\begin{aligned}
\mathsf{TV}(\widehat{x}_{N_0+1}, x_0) &\leq \mathsf{TV}(x_0, x_{t_{N_0+1}}) + \mathsf{TV}(\widehat{x}_{N_0+1}, x_{t_{N_0+1}}) \\
&\leq \mathsf{TV}(x_0, x_\delta) + \mathsf{TV}(x_T, \mathcal{N}(0, I_d)) + \sum_{n=1}^{N_0+1} \mathsf{TV}(\widehat{y}_n, x_{t_n}) \\
&\leq \mathsf{TV}(x_0, x_\delta) + \mathsf{TV}(x_T, \mathcal{N}(0, I_d)) + N_0 \cdot \left(\frac{\epsilon_{\text{sc}}}{\sqrt{L}} + \frac{\epsilon}{\min(\beta_1, \beta_2)}\right).
\end{aligned}$$

By Theorem B.9, $\mathsf{TV}(x_T, \mathcal{N}(0, I_d)) \lesssim (\sqrt{d} + \mathfrak{m}_2) \exp(-T)$. By (Lee et al., 2023, Lemma 6.4), $\mathsf{TV}(x_0, x_\delta) \leq \epsilon$. Therefore by setting $T = \Theta\left(\log\left(\frac{d \vee \mathfrak{m}_2^2}{\epsilon^2}\right)\right)$, $N_0 = \Theta\left(L \log\left(\frac{d \vee \mathfrak{m}_2^2}{\epsilon^2}\right)\right)$ and $\beta_1 = \beta_2 = \Theta\left(L \log\left(\frac{d \vee \mathfrak{m}_2^2}{\epsilon^2}\right)\right)$ in Algorithm 7 and Algorithm 8, when $\epsilon_{\mathrm{sc}} \lesssim \widetilde{\Theta}(\frac{\epsilon}{\sqrt{L}})$, we obtain $\mathsf{TV}(\widehat{x}_{N_0+1}, x_0) \lesssim \epsilon$.

The iteration complexity of Algorithm 9 given above parameters is roughly number of predictor-corrector steps times the iteration complexity in one predictor-corrector step. Note that in any corrector step and any predictor step except the last one, only $N = O(1)$ number of sub-steps are taken, therefore the iteration complexity of one predictor step (except the last step) is $\Theta(\log(\frac{\beta_1 \sqrt{d}}{\epsilon}))$ and iteration complexity of one corrector step is $\Theta(\log(\frac{\beta_2^2 d}{\epsilon^2}))$. In the last predictor step, the number of steps taken is $O\left(\log\left(\frac{1}{\delta L}\right)\right) = O(\log(L) + T)$, and thus the iteration complexity is $\Theta((\log(L) + T) \cdot \log(\frac{\beta_1 \sqrt{d}}{\epsilon}))$. We conclude that the total iteration complexity of Algorithm 9 is

$$LT \cdot \left(\Theta(\log(\frac{\beta_1 \sqrt{d}}{\epsilon})) + \Theta(\log(\frac{\beta_2^2 d}{\epsilon^2}))\right) + \Theta\left((\log(L) + T) \cdot \log\left(\frac{\beta_1 \sqrt{d}}{\epsilon}\right)\right) = \widetilde{\Theta}\left(L \log^2\left(\frac{Ld \vee \mathfrak{m}_2^2}{\epsilon}\right)\right).$$

$\square$

# D    LOG-CONCAVE SAMPLING IN TOTAL VARIATION

In this section, we give a simple proof, using our observation about trading off the time spent on the predictor and corrector steps, of an improved bound for sampling from a log-concave distribution in total variation.

**Definition D.1.** *A distribution with probability density $p$ is a log-concave distribution if $\log p$ is concave. Formally, for any $x, y$ in the domain of $p$ and for any $\lambda \in (0, 1)$, $\log f(\lambda x + (1 - \lambda)y) \geq \lambda \log f(x) + (1 - \lambda) \log f(y)$.*

Note that for this section, we assume that $\widehat{s}$ is the *true score* of the distribution and is known, as is standard in the log-concave sampling literature. We begin by recalling Shen and Lee's randomized midpoint method applied to approximate the underdamped Langevin process, for log-concave sampling in the Wasserstein metric (Shen & Lee, 2019) in Algorithm 10.

---

**Algorithm 10** RANDOMIZEDMIDPOINTMETHOD (Shen & Lee, 2019)

---

**Input parameters:**

- Starting sample $\widehat{x}_0$, Starting $v_0$, Number of steps $N$, Step size $h$, Score function $\widehat{s}$, $u = \frac{1}{L}$.

1. For $n = 0, \ldots, N - 1$:

    (a) Randomly sample $\alpha$ uniformly from $[0, 1]$.

    (b) Generate Gaussian random variable $(W_1^{(n)}, W_2^{(n)}, W_3^{(n)}) \in \mathbb{R}^{3d}$ as in Appendix A of (Shen & Lee, 2019).

    (c) Let $\widehat{x}_{n+\frac{1}{2}} = \widehat{x}_n + \frac{1}{2}\left(1 - e^{-2\alpha h}\right)v_n - \frac{1}{2}u\left(\alpha h - \frac{1}{2}\left(1 - e^{-2(h - \alpha h)}\right)\right)\widehat{s}(x_n) + \sqrt{u}W_1^{(n)}$.

    (d) Let $\widehat{x}_{n+1} = \widehat{x}_n + \frac{1}{2}\left(1 - e^{-2h}\right)v_n - \frac{1}{2}uh\left(1 - e^{-2(h - \alpha h)}\right)\widehat{s}(x_{n+\frac{1}{2}}) + \sqrt{u}W_2^{(n)}$.

    (e) Let $v_{n+1} = v_n e^{-2h} - uhe^{-2(h - \alpha h)}\widehat{s}(x_{n+\frac{1}{2}}) + 2\sqrt{u}W_3^{(n)}$.

2. Return $\widehat{x}_N$.

---

**Theorem D.2** (Theorem 3 of (Shen & Lee, 2019), restated). *Let $\widehat{s} = \nabla \ln p$, the score function of a log-concave distribution $p$ be such that $0 \preccurlyeq m \cdot I_d \preccurlyeq J_{\widehat{s}}(x) \preccurlyeq L \cdot I_d$, for the Jacobian $J_{\widehat{s}}$ of $\widehat{s}$. Let $\widehat{x}_0$ be the root of $\widehat{s}$, and $v_0 = 0$. Let $\kappa = \frac{L}{m}$ be the condition number. For any $0 < \epsilon < 1$, if we set the step size of Algorithm 10 as $h = C \min\left(\frac{\epsilon^{1/3} m^{1/6}}{d^{1/6} \kappa^{1/6}} \log^{-1/6}\left(\frac{d}{\epsilon m}\right), \frac{\epsilon^{2/3} m^{1/3}}{d^{1/3}} \log^{-1/3}\left(\frac{d}{\epsilon m}\right)\right)$*

*for some small constant $C$ and run the algorithm for $N = \frac{4\kappa}{h} \log \frac{20d}{\epsilon^2 m} \leq \widetilde{O}\left(\frac{\kappa^{7/6}d^{1/6}}{\epsilon^{1/3}m^{1/6}} + \frac{\kappa d^{1/3}}{\epsilon^{2/3}m^{1/3}}\right)$ iterations, then Algorithm 10 after $N$ iterations can generate $\widehat{x}_N$ such that*

$$W_2(\widehat{x}_N, x) \leq \epsilon$$

*where $x \sim p$.*

Now, we make the following simple observation – if we run the corrector step from Section B.2 for a short time, we can convert the above Wasserstein guarantee to a TV guarantee. We carefully trade off the time spent on the Randomized Midpoint step above and the corrector step to obtain the improved dimension dependence. Our final algorithm is given in Algorithm 11.

---

**Algorithm 11** LOGCONCAVESAMPLING (Shen & Lee, 2019)

---

**Input parameters:**

- Number of Randomized Midpoint steps $N_{\mathrm{rand}}$, Corrector steps Time $T_{\mathrm{corr}} \lesssim \frac{1}{\sqrt{L}}$, Randomized Midpoint Step size $h_{\mathrm{rand}}$, Corrector step size $h_{\mathrm{corr}}$, Score function $\widehat{s}$.

1. Let $\widehat{x}_0$ be the root of $\widehat{s}$, and let $v_0 = 0$.
2. Run Algorithm 10 with $N_{\mathrm{rand}}$ steps and step size $h_{\mathrm{rand}}$, using $\widehat{x}_0, v_0$, and let the result be $\widehat{x}'_{N_{\mathrm{rand}}}$.
3. Run Algorithm 5 starting from $\widehat{x}'_{N_{\mathrm{rand}}}$ for time $T_{\mathrm{corr}}$, using step size $h_{\mathrm{corr}}$. Let the result be $\widehat{x}_{N_{\mathrm{rand}}}$.
4. Return $\widehat{x}_{N_{\mathrm{rand}}}$.

---

We obtain the following guarantee with our improved dimension dependence of $\widetilde{O}(d^{5/12})$.

**Theorem D.3** (Log-Concave Sampling in Total Variation). *Let $\widehat{s} = \nabla \ln p$ be the score function of a log-concave distribution $p$ such that $0 \prec m \cdot I_d \preccurlyeq J_{\widehat{s}}(x) \preccurlyeq L \cdot I_d$ for the Jacobian $J_{\widehat{s}}$ of $\widehat{s}$. Let $\kappa = \frac{L}{m}$ be the condition number. For any $\epsilon < 1$, if we set $h_{rand} = C\left(\frac{\epsilon^{2/3}}{d^{5/12}\kappa^{1/3}} \log^{-1/3}\left(\frac{d\kappa}{\epsilon}\right)\right)$ for a small constant $C$, $N_{rand} = \frac{4\kappa}{h} \log \frac{20d\kappa}{\epsilon^2} \leq \widetilde{O}\left(\frac{\kappa^{4/3}d^{5/12}}{\epsilon^{2/3}}\right)$, $h_{\mathrm{corr}} = \widetilde{O}\left(\frac{\epsilon}{d^{17/36}\sqrt{L}}\right)$ and $T_{\mathrm{corr}} = O\left(\frac{1}{\sqrt{L}d^{1/18}}\right)$, we have that Algorithm 11 returns $\widehat{x}_{N_{rand}}$ with*

$$\mathsf{TV}(\widehat{x}_{N_{rand}}, x) \lesssim \epsilon$$

*for $x \sim p$. Furthemore, the total iteration complexity is $\widetilde{O}\left(d^{5/12}\left(\frac{\kappa^{4/3}}{\epsilon^{2/3}} + \frac{1}{\epsilon}\right)\right)$.*

*Proof.* By Theorem D.2, we have, for our setting of $N_{\mathrm{rand}}$ and $h_{\mathrm{rand}}$ that, at the end of step 2 of Algorithm 11,

$$W_2(\widehat{x}'_{N_{\mathrm{rand}}}, x) \leq \frac{\epsilon}{d^{1/12}\sqrt{L}}\,.$$

Then, by the first part of Corollary B.7,

$$\mathsf{TV}(\widehat{x}_{N_{\mathrm{rand}}}, x) \lesssim \epsilon + \sqrt{L}d^{17/36} \cdot \left(\frac{\epsilon}{d^{17/36}\sqrt{L}}\right) \lesssim \epsilon\,.$$

Our iteration complexity is bounded by $N_{\mathrm{rand}} + \frac{T_{\mathrm{corr}}}{h_{\mathrm{corr}}} = \widetilde{O}\left(\frac{\kappa^{4/3}d^{5/12}}{\epsilon^{2/3}} + \frac{d^{5/12}}{\epsilon}\right)$ as claimed. $\square$

## E  HELPER LEMMAS

**Lemma E.1** (Corollary 1 of (Chen et al., 2023b)). *For the ODE*

$$\mathrm{d}x_t = (x_t + \nabla \ln q_t(x_t))\,\mathrm{d}t\,,$$

*if $L \geq 1$ and $\mathbb{E}\left[\|\nabla^2 \log q_t(x)\|^2\right] \leq L$, we have, for $0 < s < t$ and $h = t - s$,*

$$\mathbb{E}\left[\|\nabla \ln q_t(x_t) - \nabla \ln q_s(x_s)\|^2\right] \lesssim L^2 dh^2 \left(L \vee \frac{1}{t}\right)\,.$$

**Lemma E.2** (Implicit in Lemma 4 of (Chen et al., 2023b)). *Suppose $L \geq 1$, $h \lesssim \frac{1}{L}$ and $t_0 - h \geq t_0/2$. For ODEs starting at $x_{t_0} = \widehat{x}_{t_0}$, where*

$$\mathrm{d}x_t = (x_t + \nabla \ln q_t(x_t))\,\mathrm{d}t$$
$$\mathrm{d}\widehat{x}_t = (x_t + \widehat{s}_{t_0}(\widehat{x}_{t_0}))\,\mathrm{d}t,$$

*we have*

$$\mathbb{E}\,\|x_{t_0-h} - \widehat{x}_{t_0-h}\|^2 \lesssim h^2 \left( L^2 dh^2 \left( L \vee \frac{1}{t_0} \right) + \epsilon_{\mathrm{sc}}^2 \right).$$

**Lemma E.3** (Lemma $B.1.$ of (Gupta et al., 2023a), restated). *Let $p_0$ be a distribution over $\mathbb{R}^d$. For $x_0 \sim p_0$, let $x_t = x_0 + z_t \sim p_t$ for $z_t \sim \mathcal{N}(0, tI_d)$ independent of $x_0$. Then,*

$$\frac{p_t(x_t + \epsilon)}{p_t(x_t)} = \mathop{\mathbb{E}}_{z_t|x_t} [e^{\frac{\epsilon^T z_t}{t} - \frac{\|\epsilon\|^2}{2t}}]$$

*and*

$$\nabla \ln p_t(x_t) = \mathop{\mathbb{E}}_{z_t|x_t} \left[ -\frac{z_t}{t} \right]$$

**Lemma E.4.** *For $q_t(y_t) \propto p_{e^{2t}-1}(e^t y_t)$, for $z_t \sim \mathcal{N}(0, (e^{2t}-1)I_d)$, we have*

$$\nabla \ln q_t(y_t) = e^t \nabla \ln p_{e^{2t}-1}(e^t y) = e^t \mathop{\mathbb{E}}_{z_t|e^t y_t} \left[ \frac{-z_t}{e^{2t}-1} \right]$$

*Furthermore,*

$$\mathop{\mathbb{E}}_{y_t \sim q_t} \left[ \|\nabla \ln q_t(y_t)\|^2 \right] \lesssim \frac{d}{t}$$

*Proof.* The first claim is an immediate consequence of the definition of $q_t$ and Lemma E.3. For the second claim, note that

$$\begin{aligned}
\mathop{\mathbb{E}}_{y_t \sim q_t} \left[ \|\nabla \ln q_t(y_t)\|^2 \right] &= e^{2t} \mathop{\mathbb{E}}_{y_t \sim q_t} \left[ \left\| \mathop{\mathbb{E}}_{z_t|e^t y_t} \left[ \frac{-z_t}{e^{2t}-1} \right] \right\|^2 \right] \\
&\leq e^{2t} \mathop{\mathbb{E}}_{y_t \sim q_t} \left[ \mathop{\mathbb{E}}_{z_t|e^t y_t} \left[ \frac{\|z_t\|^2}{(e^{2t}-1)^2} \right] \right] \\
&= e^{2t} \mathop{\mathbb{E}}_{z_t} \left[ \frac{\|z_t\|^2}{(e^{2t}-1)^2} \right] \\
&= \frac{e^{2t} \cdot d}{e^{2t}-1} \quad \text{since } z_t \sim \mathcal{N}(0, (e^{2t}-1)I_d) \\
&\lesssim \frac{d}{t}.
\end{aligned}$$
$\qquad\qquad\square$

