# OpenReview forum: "Faster Diffusion Sampling with Randomized Midpoints: Sequential and Parallel"
_ICLR.cc/2025/Conference — ICLR 2025 Poster_

### Official Review · Reviewer_g31r · 2024-11-01

**Soundness:** 3
**Presentation:** 2
**Contribution:** 3
**Rating:** 6
**Confidence:** 4

**Summary:**

This paper studies diffusion models and introduces a faster sampling algorithm utilizing the randomized midpoint method within a predictor-corrector framework. In the predictor step, the paper introduces the randomized midpoint method, which allows for bounding the squared displacement into “bias” and“variance”terms. The corrector step follows the analysis in Chen et al.,2023b. This approach reduces discretization error, leading to an algorithm with improved theoretical dependency on the dimension $d$. Similar analysis can also be used in the setting of log-concave sampling in total variation distance, achieving a better dependency in $d$.

**Strengths:**

1. The organization is clear, highlighting the main contribution.

2. The analysis method is very novel, and it’s an interesting result to see that the dimensional dependency for sampling can be improved to be better than $\tilde O(\sqrt{d})$.

3. The paper shows that the analysis can also be extended to parallel algorithms.

**Weaknesses:**

1.The assumptions are a little strong.  It has been shown in [1][2] that the assumption of Lipschitz score can be relaxed. Moreover, it is often enough to assume that the true score is Lipschitz (Assumption 2.2). However, in this paper, the estimated score is also required to be Lipschitz. Although I don’t think it’s a big problem, can the authors remark about where this additional assumption is used?

2.There are issues with the proof of Lemma A.2. The argument in (Line 736 equation 17) holds only under pointwise estimation (the estimated score is accurate for any $x$). However, in Assumption 2.4, it is defined as the expectation. Similar conditions occur in Line 755. As all the lemmas after this depend on Lemma A.2, the main result is correct only with a stronger assumption, which is not reflected in the paper.

3.It misses some background knowledge and discussion:

(1) The paper discusses log-concave sampling in section 3.6. However, there is no background knowledge in this paper. It is unclear what is the exact setting. Thus, the analysis in Appendix C is confusing.

(2) In the main content, all the theorems are informal. Therefore, in the proof technique sections, like Sections 3.3, and 3.4, the target of the argument is ambiguous. I think at least a formal version of the theorem should be provided, and more details of the proof should be provided to help readers understand.

4.Some typos: I think Algorithms 1, 2, and 3 are exactly the same as Algorithms 4,5,6 in the appendix. Thus, the reference in Algorithm 3 Line 3 should be Algorithm 1 instead of Algorithm 4, right?

I'm willing to increase my score if the authors can solve my concerns, especially weakness 2.

----
[1] Benton et al. 2024, Nearly $d$-Linear Convergence Bounds for Diffusion Models via Stochastic Localization, ICLR2024

[2] Chen et al. 2023, Improved Analysis of Score-based Generative Modeling: User-Friendly Bounds under Minimal Smoothness Assumptions, ICML2023

**Questions:**

1. As claimed in [1], the linear dependency on $d$ is optimal for KL divergence, thus $\sqrt{d}$ for TV. Can you remark on this compared with the better than $\sqrt{d}$ result in this paper?

2. In the proof of Theorem A.10, given the value of the hyperparameters in Line 1109, can you explain more about how you can get the iteration complexity? Is it max $\\{1/h_{pred}, 1/h_{corr}\\}$? Then why the dependency on $d$ is not $d^{17/36}$, which is larger than $d^{5/12}$?

---

> ### Author Response · Authors · 2024-11-19
>
> We thank the reviewer for the review and helpful feedback. We respond to your specific concerns and questions below.
>
> **Re: Lipschitz assumption on estimated score:** We need this assumption in the proofs so that we get the property ``when the position $\hat x$ is not too far off from the true position $x$, the estimated score is also not too far off". For instance, this property is used in Lemma A.2.
>
>
> **Re: proof of Lemma A.2:** You are absolutely right and we apologize for the sloppiness in this proof. Our proof is meant to work with our Assumption 2.4 as stated, and not the stronger point-wise approximation. Specifically, what we meant to say in Line 736 is that 1) at the true position $x$, by Assumption 2.4, the expected difference of the estimated score and true score is bounded; and 2) our estimated $\hat x$ is not too far off from the true $x$, and due to lipschitz continuity assumption on both the estimated score $s$ and true score $\nabla q$, both the estimation score and the true score evaluated at the estimated position $\hat x$ is close to their value evaluated at the true position $x$. However, in this part of the proof, we got confused in the writing, and instead of applying Assumption 2.2, 2.3 and 2.4 together, we applied Assumption 2.2 and the point-wise version of Assumption 2.4. We apologize again for this mistake and have uploaded a corrected version of our submission. We mainly changed Lemma A.2 and Lemma B.2, but also parts of other theorems where these Lemmas are used, the main flow of our argument remain unchanged.
>
> **Re: background discussion:** We added a definition of the log concave function in section C, and will continue to work on improving the clarity of our discussion for the next version of our paper.
>
> **Re: optimality of $d$ dependence for KL and TV:**  A crucial difference between [1] and our work is that [1] mainly analyze the DDPM setting with SDE, while we analyze the DDIM setting with probability flow ODE. Also, to the best of our knowledge, although the iteration complexity in [1] has a linear dependence on $d$, their optimality discussion does not rule out sublinear (or better than $\sqrt{d}$) iteration complexity. Specifically, in their calculation, the KL error induced by discretizing time is of order $\tilde{O}(d h)$, where $h$ is the step size. They mentioned that in the KL error bound, the linear $d$ dependence in this discretization error bound is optimal, however, the dependence on the step size $h$ is not necessarily optimal and [1] theorized that this could be improved to $O(d h^\beta)$ for $\beta \geq 2$, especially when there is a smoothness assumption for the score function. In this case, one can set the step size to be $h = \Theta(d^{-1/\beta})$ and obtain $d^{1/\beta}$ iteration complexity.
>
> **Re: iteration complexity in A.10:** We are actually running predictor and corrector step for different amount of time. We are running each predictor step for $1/L$ time (specified in Algorithm 3), but we run each corrector step for $T_{corr} = \Theta(L^{-1/2}d^{-1/18})$ time (as specified in Theorem A.10). Since we take corrector step size $h_{corr} = \Theta(d^{-17/36})$, we need $\Theta(d^{5/12})$ iteration complexity to complete one corrector step, which coincides with the iteration complexity to complete one predictor step.

---

> > ### Comment · Reviewer_g31r · 2024-11-20
> > **Response to the authors**
> >
> > Thanks for your response, which addresses my concerns.
> >
> > After reading the corrected version, I find it to be sound. One of my remaining concerns is that Assumption 2.4 is still a little strong since it requires the condition to hold for any $t$. A more standard assumption might involve the expectation over $t$. I do not consider it a huge problem, though.  Therefore, I have raised my scores.
> >
> > Additionally, I recommend that the authors carefully review the paper to ensure there are no similar issues remaining. And the presentation can be further improved.

---

### Official Review · Reviewer_henE · 2024-11-02

**Soundness:** 4
**Presentation:** 3
**Contribution:** 3
**Rating:** 8
**Confidence:** 3

**Summary:**

This paper uses the randomized midpoint method to solve the reverse-diffusion SDE in generative diffusion models, to generate samples.  The randomized midpoint method is a more efficient method of solving SDEs than methods which incur a first-order error, such as the Euler-Maruyama integrator, as long as the coefficients are smooth functions.   The randomized midpoint method was originally developed for implementing the Hamiltonian Monte Carlo Markov Chain algorithm, but the here the authors apply the same technique in the setting of generative diffusion models.  This allows the authors to obtain faster runtime bounds for sampling (in the total variation norm) from the reverse-diffusion SDE in diffusion models, provided the certain smoothness assumptions are satisfied.  As a by-product of their analysis the authors also obtain improved bounds for sampling from log-concave distributions using the Langevin MCMC algorithm.

**Strengths:**

The paper obtains guarantees, in the TV norm, for solving the reverse SDE in diffusion models, with improved dimension dependence on the runtime.

The paper obtains improved guarantees, which are logarithmic in dimension, for parallelized solving of reverse SDE in diffusion models.

As a by-product of their analysis the authors also obtain improved bounds for sampling from log-concave distributions using the Langevin MCMC algorithm.

The paper introduces a new technique (the randomized midpoint method, originally developed for log-concave sampling via MCMC algorithms) to the area of generative diffusion models.

**Weaknesses:**

The paper does not include any numerical simulations.  Numerical simulations would help evaluate whether their algorithm works well in practice (both in terms of runtime and in terms of sample generation quality).

Also, while the authors adapt a new technique to the generative diffusion models, this is still an existing technique, but was used in a different area (MCMC for log-concave sampling).  That being said, I still feel the technical contribution is good since the authors introduce an existing technique to a new area.

**Questions:**

Can the authors provide simulations of their methods? It would be interesting to see if their method works in practice.  This concerns both the runtime and the quality of the generated samples.

---

> ### Comment · Reviewer_henE · 2024-11-26
>
> I am assuming the authors have not yet had the time to respond to my review.  I will reply once they respond.

---

> > ### Author Response · Authors · 2024-11-26
> >
> > We apologize for the delay! Our simulations for some larger datasets are still in the process of being generated (our simulation on CIFAR-10 has finished). We will respond soon with a description and plots of our simulation results.

---

> ### Author Response · Authors · 2024-11-28
>
> We thank the reviewer for the review and helpful feedback. We respond to your specific concerns and questions below.
>
> **Re: Experiments:** We have added an experimental section in our updated pdf (Appendix A), comparing the performance of our proposed randomized midpoint technique with the standard DDIM scheduler. Our experiments generate images using the two schedulers on two diffusion models -- one trained on CIFAR-10 images, and the other on CelebAHQ images. We show that, as predicted by our theory, randomized midpoint consistently outperforms DDIM in terms of FID scores, which measure the quality of generated samples relative to the training dataset.

---

> > ### Comment · Reviewer_henE · 2024-12-01
> >
> > Thank you for the simulations.  The simulations seem to be showing your method is faster than another method called "DDIM", so that is good.  However, it is a bit unclear what is the "default DDIM scheduler" method you are comparing to?  Which paper is DDIM, and what does "DDIM" stand for?  It's difficult to evaluate the significance of your simulations without knowing what is the baseline paper you are comparing to.

---

> > > ### Author Response · Authors · 2024-12-01
> > >
> > > Hi, thanks for your response.
> > >
> > > DDIM refers to the standard Euler discretization of the ODE, as described in this paper: https://arxiv.org/abs/2010.02502

---

> > > > ### Comment · Reviewer_henE · 2024-12-01
> > > >
> > > > Thanks-- I have raised my score to reflect the added simulations.

---

### Official Review · Reviewer_u9VU · 2024-11-02

**Soundness:** 3
**Presentation:** 3
**Contribution:** 3
**Rating:** 6
**Confidence:** 2

**Summary:**

The manuscript introduces a novel approach for enhancing sampling efficiency in diffusion models by incorporating a randomized midpoint method. This method, inspired by Shen and Lee's work on log-concave sampling, is adapted to diffusion processes to reduce sample complexity and improve iteration bounds. The authors present theoretical guarantees for both sequential and parallel sampling, addressing the limitations of previous diffusion model sampling techniques.

**Strengths:**

**Theoretical Innovation.** The manuscript extends a randomized midpoint technique, traditionally used in other domains, to diffusion sampling, presenting significant advancements in sample complexity for smooth distributions.

**Parallel Sampling Framework.** Including a parallel sampling approach with provable efficiency bounds is particularly notable, potentially paving the way for scalable implementations in large-scale generative modeling tasks.

**Solid Analysis.** The authors offer a rigorous mathematical foundation for their method, with thorough proofs and well-defined assumptions on score functions and dimensionality dependence.

**Weaknesses:**

**Assumptions and Practical Applicability.** While the theoretical advancements are clear, the reliance on strong Lipschitz assumptions and high accuracy in score estimates might limit practical applicability. The authors could consider discussing the feasibility of these assumptions in real-world diffusion models.

**Lack of Experimental Validation.** Despite the paper's focus on theoretical results, experimental validation would have been beneficial. Empirical benchmarks could highlight practical gains and offer insights into scenarios where the proposed method outperforms existing approaches.

**Parallelization Overhead.** While the approach achieves polylogarithmic round complexity, it requires significant parallel resources to fully realize these benefits. The paper could improve by addressing potential bottlenecks and clarifying hardware implications.

**Questions:**

**Space Complexity and Scalability.** Can the authors elaborate on how the proposed method would scale with large data distributions in practical applications? How feasible is this method for scenarios requiring substantial memory or computation resources?

**Empirical Benchmarks.** Could the authors provide insights on the types of empirical tests they recommend or any preliminary results? Testing on standard datasets or benchmarks in score-based generative models would be particularly valuable for evaluating this method’s efficiency.

---

> ### Author Response · Authors · 2024-11-28
>
> We thank the reviewer for the review and helpful feedback. We respond to your specific concerns and questions below.
>
> **Re: Experiments:** We have added an experimental section in our updated pdf (Appendix A), comparing the performance of our proposed randomized midpoint technique with the standard DDIM scheduler. Our experiments generate images using the two schedulers on two diffusion models -- one trained on CIFAR-10 images, and the other on CelebAHQ images. We show that, as predicted by our theory, randomized midpoint consistently outperforms DDIM in terms of FID scores, which measure the quality of generated samples relative to the training dataset.
>
> **Re: Space Complexity and Scalability:**
> For the sequential setting, the space complexity is no different than the space complexity of standard schedulers such as DDIM. For the parallel setting, a recent work [1] has shown that an algorithm similar to a parallelized version of DDIM performs well in practice, provided a sliding window method is used to maintain samples $x_{t:t+p}$ for window size $p$, to reduce memory usage. We expect that the same approach would work to implement our randomized-midpoint-based parallel sampling algorithm.
>
> [1]: https://arxiv.org/pdf/2305.16317

---

### Official Review · Reviewer_s9Bu · 2024-11-04

**Soundness:** 3
**Presentation:** 3
**Contribution:** 3
**Rating:** 6
**Confidence:** 4

**Summary:**

The authors explore sampling for diffusion with mild assumptions and introduce both sequential and parallel algorithms, which extend the ODE-based predictor-corrector framework initially presented in [CCL+23a]. In their sequential approach, they adeptly utilize the randomized midpoint method in[SL19] to discretize the probability flow ODE during the predictor phase, and use the corrector method outlined in [CCL+23a] with a short time $T$. Impressively, this method achieves a complexity of $O(L^{5/3}d^{4/9}/\varepsilon)$, which marks a substantial improvement over the previous best complexity of $O(L^2d^{1/2}/\varepsilon)$. For the parallel algorithm, the authors also apply a similar parallel randomized midpoint method from [SL19] for the predictor phase and use the parallel underdamped Langevin method from [ACV24] as the corrector. This innovative combination gives the first provable guarantees for parallel sampling within diffusion models. Furthemore, this paper gives an improved bound for log-concave sampling.

**Strengths:**

- Their method achieves significantly better sequential complexity and parallel complexity.

- Overall I found the paper well-written and clear.

**Weaknesses:**

- Their sequential result requires a slightly stronger assumption on the score estimation error, but I think it is fine.

- The technical contributions are limited to combining existing ODE-based predictor-corrector framework and discrete schemes.

**Questions:**

- is it possible to improve the total query complexity of paralell version to $d^{4/9}$?

- What are the primary challenges associated with removing the smoothness assumption?

- In the parallel version of sampling for diffusion models and the log-concave sampling algorithm, the predictor-corrector framework runs only once, whereas in the sequential version, the predictor-corrector loop is repeated $N_0$ times. Why does this difference occur?

---

> ### Author Response · Authors · 2024-11-21
>
> We thank the reviewer for the review and helpful feedback. We respond to your specific concerns and questions below.
>
> **Re: parallel query complexity:** We believe that at least within our proof framework, there is a fundamental tradeoff for parallel algorithms between iteration complexity and total query complexity. Intuitively, randomized midpoint method achieves reduced query complexity by reducing the bias of estimation in each step, and this benefit accumulates as number of iteration grows. For instance, a generalized version of our parallel algorithms can achieve $d^{1/2 - \epsilon}$ (for $\epsilon \leq 1/12$) query complexity with $\log^2 d \cdot d^{\Theta(\epsilon)}$ iteration complexity. In our proof, we set $\epsilon = 0$ to achieve $\mathsf{polylog}(d)$ iteration complexity.
>
> **Re: smoothness assumption:** Typically, when the smoothness assumption is not present, prior works bound the "effective" smoothness by a function of $\sigma_t$ and $d$. However, the existing bounds ([1], [2]) for this effective smoothness are not strong enough to even achieve a sublinear in $d$ bound on the iteration complexity, and we run into similar barriers here. Within the diffusion community, achieving such a sublinear bound is a well-known question, and is orthogonal to our focus here.
>
> **Re: predictor-corrector loop:** In both our sequential and parallel algorithm, the predictor-corrector loop runs for multiple ($\tilde{O}(1)$) times. (Note that Algorithm 3 for sequential loops and Algorithm 9 for parallel loops looks almost identical.) However, it is true that in the parallel algorithm, we only take $\tilde{O}(1)$ predictor steps in one predictor-corrector loop; while in the sequential algorithm, we take $\tilde{O}(d^{5/12})$ predictor steps. In the parallel algorithm, we are able to take larger step sizes since we update many randomized midpoints in parallel in one predictor step.
>
> [1] Benton et al. 2024, Nearly-Linear Convergence Bounds for Diffusion Models via Stochastic Localization, ICLR2024

---

> > ### Comment · Reviewer_s9Bu · 2024-11-25
> >
> > Thanks for your response, which addresses my concerns. I choose to retain the current score.

---

### Meta-Review · Area_Chair_pecS · 2024-12-22

**Metareview:**

This paper considers diffusion sampling via the randomized midpoint method for log-concave sampling. Authors improve the existing dimension dependence for sampling from arbitrary smooth distributions in total variation distance. Authors also consider a parallel version of this algorithm and provide respective guarantees.


This paper was reviewed by three expert reviewers and received the following Scores/Confidence: 6/2, 8/3, 6/4, 6/4. Even though certain results are not surprising, I still think the paper is studying an interesting topic and the results are relevant to ICLR community. The following concerns were brought up by the reviewers:

- The idea is straightforward, and the technical contribution seems limited. Authors should explain the novelty in their work more clearly.

- The feasibility of the conditions should be discussed in detail.

- Authors discuss a 'new' intergrator and provide improved guarantees for it. I expected to see extensive numerical results that validates the improved performance. I suggest authors to extend their experiments in camera ready.


Authors should carefully go over reviewers' suggestions and address any remaining concerns in their final revision. Based on the reviewers' suggestion, as well as my own assessment of the paper, I recommend including this paper to the ICLR 2025 program.

**Additional Comments On Reviewer Discussion:**

Authors successfully addressed majority of the concerns raised by the reviwerers.

---

### Decision · Program_Chairs · 2025-01-22

Accept (Poster)